# TNF antagonist sensitizes synovial fibroblasts to ferroptotic cell death in collagen-induced arthritis mouse models

Jiao Wu [1,2,3,9 ✉], Zhuan Feng[2,9], Liang Chen[4,9], Yong Li[5,6,9], Huijie Bian[2], Jiejie Geng[2], Zhao-Hui Zheng[1], Xianghui Fu[1], Zhuo Pei[2], Yifei Qin[2,7], Liu Yang[2], Yilin Zhao[8], Ke Wang[2], Ruo Chen[2], Qian He[2], Gang Nan[2], Xuejun Jiang [3 ✉], Zhi-Nan Chen [2 ✉] & Ping Zhu [1 ✉]

Ferroptosis is a nonapoptotic cell death process that requires cellular iron and the accumulation of lipid peroxides. In progressive rheumatoid arthritis (RA), synovial fibroblasts proliferate abnormally in the presence of reactive oxygen species (ROS) and elevated lipid oxidation. Here we show, using a collagen-induced arthritis (CIA) mouse model, that imidazole ketone erastin (IKE), a ferroptosis inducer, decreases fibroblast numbers in the synovium. Data from single-cell RNA sequencing further identify two groups of fibroblasts that have distinct susceptibility to IKE-induced ferroptosis, with the ferroptosis-resistant fibroblasts associated with an increased TNF-related transcriptome. Mechanistically, TNF signaling promotes cystine uptake and biosynthesis of glutathione (GSH) to protect fibroblasts from ferroptosis. Lastly, low dose IKE together with etanercept, a TNF antagonist, induce ferroptosis in fibroblasts and attenuate arthritis progression in the CIA model. Our results thus imply that the combination of TNF inhibitors and ferroptosis inducers may serve as a potential candidate for RA therapy.

[1] Department of Clinical Immunology, Xijing Hospital, Fourth Military Medical University, Xi'an, China. [2] National Translational Science Center for Molecular Medicine & Department of Cell Biology, Fourth Military Medical University, Xi'an, China. [3] Cell Biology Program, Memorial Sloan-Kettering Cancer Center, New York, NY, USA. [4] School of Medicine, Shanghai University, Shanghai, China. [5] Xijing 986 Hospital Department, Fourth Military Medical University, Xi'an, China. [6] The Second Affiliated Hospital of Xi'an Jiaotong University, Xi'an, China. [7] School of Pharmacy, Guangdong Pharmaceutical University, Guangzhou, China. [8] Department of Oncology, Xijing Hospital, Fourth Military Medical University, Xi'an, China. [9] These authors contributed equally: Jiao Wu, Zhuan Feng, Liang Chen, Yong Li. ✉email: jiaowubio@hotmail.com; jiangx@mskcc.org; znchen@fmmu.edu.cn; zhuping@fmmu.edu.cn

Rheumatoid arthritis (RA) is an autoimmune inflammatory disorder during which a hyperplastic rheumatoid synovium overgrows the underlying cartilage surface and invades the cartilage and bone, leading to progressive joint destruction[1,2].

Inflammation and disease progression can be efficiently controlled by biological therapies targeting cytokines such as TNF and IL-6, which contribute to the joint destruction and inflammation of RA[3]. However, current therapies are unable to completely prevent progressive joint erosion and cure this disease. Furthermore, these treatments share a similar therapeutic response ceiling[4,5]. Therefore, novel therapies that can overcome the persistence of inflammation in RA need to be developed.

Ferroptosis is programmed necrosis that arises from the accumulation of lipid peroxides triggered by aberrant metabolic and biochemical processes[6–8]. Although the physiological function of ferroptosis remains to be unambiguously demonstrated, ferroptosis has been shown to be a significant cause of various pathological conditions, including ischemic organ injury and neurodegeneration, and a potential therapeutic strategy for cancer through monotherapy or as a component of combination therapies. Recent studies have shown that ferroptosis may be involved in innate immunity and thus play a role in regulating inflammatory damage, signal transduction, and cell growth[9,10]. However, knowledge regarding the precise crosstalk between the inflammatory process and ferroptosis remains incomplete.

As one of the dominant cell types in hyperplastic tissue, activated synovial fibroblasts contribute to inflammation, angiogenesis, and matrix degradation by producing inflammatory cytokines, proangiogenic factors, and matrix-degrading enzymes[11]. Owing to the dominant role of synovial fibroblasts in RA pathophysiology, new therapeutic approaches targeting activated synovial fibroblasts are of central interest[12]. Therapies targeting cytokines reduce the destructive potential of fibroblasts but cannot suppress permanent fibroblast activation[13]. Highly reactive oxygen free radicals are believed to be involved in joint inflammation and destruction[14], and synovial fibroblasts proliferate abnormally under oxidative stress[14,15] (which contributes to inflammation and joint damage), indicating that synovial fibroblasts are protected from oxidative stress in a proinflammatory environment. Thus, intervention in such protective signaling may be used as a new strategy against fibroblast activation.

In this study, we investigate how synovial fibroblasts survive in response to physiological ferroptotic stress in the hyperplastic rheumatoid synovium and synovial fluid of RA patients. TNF, which is a crucial proinflammatory cytokine in RA pathogenesis, can inhibit the onset of ferroptosis by upregulating solute carrier family 7 member 11 (SLC7A11), glutamate-cysteine ligase catalytic subunit (GCLM), and glutamate-cysteine ligase regulatory subunit (GCLC), thus promoting cystine uptake and the biosynthesis of cellular GSH. The combination of a TNF antagonist and a low dose of ferroptosis inducer induces ferroptosis in synovial fibroblasts and markedly attenuates arthritis progression in the CIA model. These findings suggest a mechanism by which TNF regulates ferroptosis resistance and implicate the therapeutic potential of ferroptosis-based therapies targeting dysregulated fibroblasts in a wide range of diseases.

## Results

### The hyperplastic rheumatoid synovium and the synovial fluid of RA patients show increased levels of lipid peroxidation and iron. Despite the mechanistic and phenotypic differences between RA and osteoarthritis (OA), OA is often used as "disease" control to reveal the autoimmune characteristics in RA. We first measured several putative biomarkers of oxidation and lipid peroxidation in RA and osteoarthritis (OA), including the 8-hydroxy-2′-deoxyguanosine (8-OHdG) oxidative product[16], 4-hydroxynonenal (4-

HNE) modifications, and malondialdehyde (MDA), which are two main byproducts of lipid peroxidation[17]. The hyperplastic rheumatoid synovium appears to show higher levels of 8-OHdG (Fig. 1a) and 4-HNE (Fig. 1b) compared with that of OA patients. However, no significant difference in the staining of 8-OHdG and 4-HNE was found between the patients with high and moderate disease activity (Supplementary Fig. 1a–c). In RA, both the synovial lining layer and sublining layer undergo expansion and enrichment of fibroblasts. The staining of 4-HNE remained unchanged in the fibroblasts of the lining layer (LL, VCAM-1$^+$) and sub-lining layer (SL, CD248$^+$) (Fig. 1c, Supplementary Fig. 1d). In inflamed joints of mice with collagen-induced arthritis (CIA), the synovium also showed stronger staining for 8-OHdG and 4-HNE than the joints of normal mice (Fig. 1a, b). In the synovial fluid of RA patients, the levels of MDA were increased with advanced disease activity, while the levels of 8-OHdG were not affected (Fig. 1d, Supplementary Fig. 1e). In addition to elevated lipid peroxidation formation, we observed much higher iron levels in the synovial fluid of patients with high disease activity than in that of patients with moderate disease activity (Fig. 1e). These data confirmed the accumulation of lipid peroxides and iron overload in the hyperplastic rheumatoid synovium and synovial fluid of RA patients.

### The ferroptosis inducer displays significant resolution for joint inflammation and destruction in an established CIA model. The excessive production of ROS can serve as important intracellular signaling molecules that amplify the inflammatory–proliferative response in RA synovium[18]. To determine the role of lipid peroxidation in the development of arthritis, we treated mice with an established CIA experimental arthritis model with imidazole ketone erastin (IKE, 40 mg/kg daily), a metabolically stable analog of erastin that induces lipid peroxidation bursts and ferroptosis[19], in the presence or absence of liproxstatin-1 (10 mg/kg every 2 days), a ferroptosis inhibitor that attenuates the accumulation of lipid ROS (Supplementary Fig. 1f). Surprisingly, although oxidative stress is believed to be a contributing factor in the pathogenesis of RA, IKE attenuated the severity of synovial inflammation and accelerated resolution when active inflammation was present, whereas liproxstatin-1 had little effect on joint swelling and inflammation (Fig. 1f, g). IKE also led to reduced cartilage and bone damage and pannus formation (Fig. 1h–j). In contrast, liproxstatin-1 treatment started at the pre-symptomatic stages of CIA (at Day 7 after the first immunization) continuously prevented the development of joint inflammation (Supplementary Fig. 1g–i) and joint destruction (Supplementary Fig. 1j).

### The ferroptosis inducer decreases fibroblast populations in the CIA synovium. We then found that the mitigation of CIA symptoms caused by IKE was accompanied by a marked induction of PTGS2 expression and a reduction in GPX4 levels (markers of oxidative stress and ferroptosis) in the remaining synovial tissue, corroborating the effect of IKE on inducing ferroptosis in vivo (Supplementary Fig. 1k). Fibroblast activation protein-α (FAPα) is a widely accepted marker for fibroblasts in RA; these cells are located in both the inner and outer layers of the synovium[20]. The number of FAPα+ fibroblasts in the synovium of RA patients significantly outnumbered that in OA patients (Fig. 2a). Interestingly, we found that synovial FAPα+ fibroblasts were almost undetectable under noninflammatory conditions but increased in the inflamed synovium of mice with CIA (Fig. 2b). IKE treatment strikingly decreased the population of FAPα+ fibroblasts (Fig. 2b). Although IKE seemed to increase the mean fluorescence intensity of F4/80+ macrophages, the Cohen's $d$ was only 0.21, which indicates a small effect size. To assess whether fibroblasts are more sensitive to ferroptosis

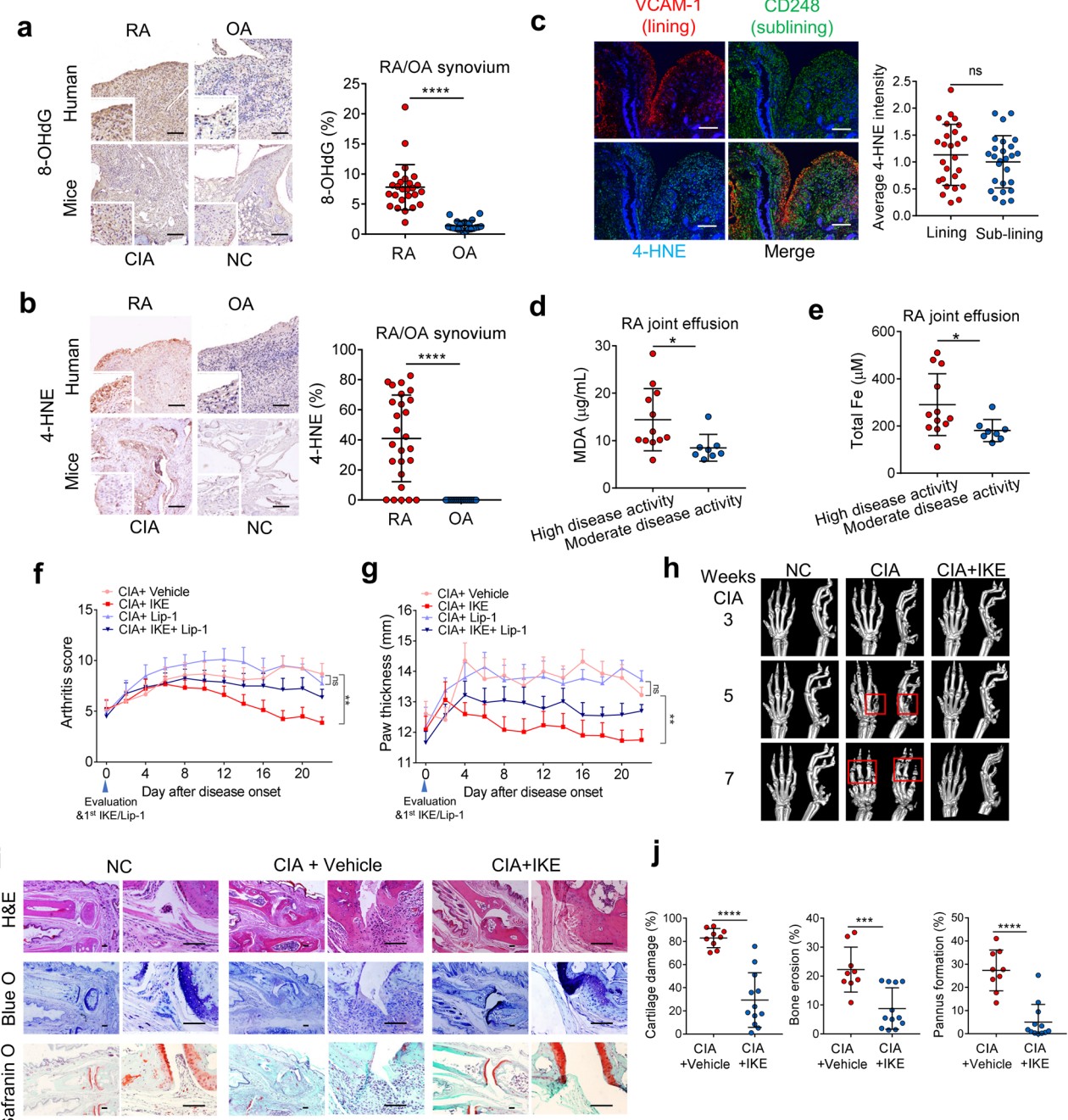

induction than other cell types in hyperplastic synovium, we treated disaggregated single cells isolated from synovial biopsies of RA patients with the GPX4 inhibitor RSL3. As shown in Fig. 2c and Supplementary Fig. 2, short-term RSL3 treatment resulted in increased cell death in FAPα+ fibroblasts but not in CD68+ macrophages, CD31+ endothelial cells, CD3+ T cells or CD19+ B cells. Next, we isolated circulating fibrocytes from peripheral blood mononuclear cells (PBMCs) and fibroblasts from the synovial fluid of RA patients (Supplementary Fig. 3) and found that synovial fibroblasts from the synovial fluid were more sensitive to RSL3 than circulating fibrocytes (Fig. 2d, e), suggesting that the inflamed joint is inclined to promote ferroptosis and lipid ROS production. This is in line with the observation that the joints of patients with high disease activity show more lipid peroxidation and iron levels than those of patients with lower disease activity. Collectively, these data suggest that ferroptosis

induction could effectively decrease fibroblast populations in the synovium of mice with CIA, thereby mitigating inflammation and tissue damage. Interestingly, the number of F4/80+ macrophages within the 50 μm radium of surviving FAPα+ fibroblasts that evaded IKE-induced ferroptosis significantly increased in the joints of IKE-treated CIA mice (Fig. 2f, g), giving rise to the assumption that macrophages may protect fibroblasts against lipid oxidative stress and ferroptotic cell death.

**Single-cell RNA sequencing detects fibroblast subsets with distinct susceptibility to ferroptosis and diversification of functional roles in the synovium of an arthritis model.** To investigate the existence of synovial fibroblast subsets with distinctive susceptibility to ferroptosis in the synovium of arthritis model, we analyzed the single-cell RNA sequencing data of CD45− non-hematopoietic cells from the synovium of mice subjected to an

**Fig. 1 The ferroptosis inducer IKE modulated joint inflammation and tissue damage. a** Left, representative immunohistochemical staining of 8-OHdG in hyperplastic rheumatoid synovium of rheumatoid arthritis (RA) and osteoarthritis (OA) patients and in the inflamed joint tissue of collagen-induced arthritis (CIA) model mice. Scale bars, 100 μm. Right, Quantitative comparison of 8-OHdG in RA and OA patients ($n = 26$ RA patients; $n = 21$ OA patients). ****$P < 0.0001$ (two-tailed $t$-test). **b** Left, representative immunohistochemical staining of 4-HNE in hyperplastic rheumatoid synovium of RA and OA patients and in the inflamed joint tissue of CIA model mice. Scale bars, 100 μm. Right, Quantitative comparison of 4-HNE in RA and OA patients ($n = 26$ RA patients; $n = 21$ OA patients). ****$P < 0.0001$ (two-tailed $t$-test). **c** Representative fluorescent multiplex IHC staining (left panel) and quantification (right panel) of RA joint synovium labeled with anti-VCAM-1 (red), anti-CD248 (green), anti-4-HNE (light blue), and DAPI (blue). Scale bars, 200 μm. ns, $P = 0.3804$ (two-tailed $t$-test). **d** MDA concentration in the joint fluid of RA patients with different disease activities. *$P = 0.0274$ (two-tailed $t$-test). **e** Total iron concentrations in the joint fluid of RA patients. *$P = 0.0365$ (two-tailed $t$-test). High disease activity, $n = 12$ RA patients; moderate disease activity, $n = 8$ RA patients (**d** and **e**). Joint inflammation was measured by arthritis score (**f**) (**$P = 0.0011$; ns, $P = 0.8455$; one-way ANOVA followed by multiple comparisons was performed to compare the means of arthritis score at the end point) and paw thickness (**g**) (**$P = 0.0037$; ns, $P = 0.5670$; one-way ANOVA followed by multiple comparisons) in CIA mice intraperitoneally injected with 40 mg/kg IKE every day and/or 10 mg/kg liproxstain-1 (Lip-1) every 2 days for 22 days. $n = 9$ mice for CIA + Vehicle and CIA + IKE group, $n = 8$ mice for CIA + Lip-1 and CIA + IKE + Lip-1 group. **h** Representative micro-computed tomography (micro-CT) images of control and CIA model mice with or without IKE treatment. **i** Images of hematoxylin and eosin (H&E), toluidine blue O, and safranin O staining of representative joints in control and CIA mice with or without IKE treatment at day 22 after treatment initiation. Scale bars, 100 μm. **j** Quantification of the histomorphometric analysis of cartilage damage, bone erosion, and pannus formation. ****$P < 0.0001$, ***$P = 0.0006$; two-tailed $t$-test. $n = 9$ joints for CIA group, $n = 12$ joints for CIA + IKE group. Data in **f** and **g** are presented as mean ± SEM. Other data are presented as mean ± SD. Source data are provided as a Source data file.

arthritis model[20]. We used t-distributed stochastic neighbor embedding (tSNE) visualization of the cells and identified 18 major clusters, 9 of which were fibroblast populations (Fig. 3a). Based on previous studies revealing the function of ferroptosis-related genes that regulate iron metabolism, lipid metabolism, and oxidant metabolism[21], we tried to predict the susceptibility of different clusters to ferroptosis induction (Fig. 3b, c). Multicolor immunofluorescence analyses revealed that the most significant marker genes identified (ferroptosis-sensitive fibroblast subset: Mfap4, ferroptosis-resistant fibroblast subset: Sparcl1) could distinguish different subsets among FAPα+ fibroblasts in the synovium of RA patients and CIA mice (Supplementary Fig. 4, Fig. 3d, e). More Mfap4-positive, ferroptosis-sensitive fibroblasts than Sparcl1-positive, ferroptosis-resistant fibroblasts were scored in hyperplastic tissue that invaded the cartilage in comparison with noninvasive hyperplastic tissue (Fig. 3f). To confirm the distinct sensitivity of the subsets to ferroptosis, we treated CIA mice with a low dose of IKE (20 mg/kg, twice/week). The staining of Mfap4 in surviving FAPα+ synovial fibroblasts of IKE-treated mice was markedly decreased compared to that of untreated CIA mice, whereas the staining of Sparcl1 was unchanged after IKE treatment (Fig. 3g). Gene set enrichment analysis (GSEA) suggested a diversification of functional roles between the two subsets. The predicted ferroptosis-sensitive subset showed a significant enrichment of genes in categories related to extracellular matrix formation and remodeling, while the predicted ferroptosis-resistant subset was strongly characterized by the expression of genes involved in proliferation and the cell cycle (Fig. 3h, i).

To determine the mechanisms responsible for the observed regulation of ferroptosis sensitivity in synovial fibroblasts, we performed Kyoto Encyclopedia of Genes and Genomes (KEGG) analysis and Gene Ontology (GO) analysis (Supplementary Figs. 5 and 6). We identified the tumor necrosis factor (TNF) α pathway as one of the most highly enriched pathways in the ferroptosis-resistant subset (Fig. 3j). To validate the role of the activated TNF pathway for fibroblast functions, we treated synovial fibroblasts from RA patients with TNF. GSEA suggested that long-term exposure of fibroblasts to TNF increased the expression of genes related to proliferation and decreased genes related to the extracellular matrix of synovial fibroblasts from individuals with RA (Supplementary Fig. 7a–c). In contrast, TNF inhibition increased the invasion of protrusions from the main body of fibroblast spheroids (Supplementary Fig. 7d, e). In conclusion, these data support the existence of fibroblast subsets with distinct ferroptosis sensitivity in CIA synovium.

**Potential crosstalk between macrophages and fibroblast subsets through TNF signaling via TNFRSF1A and TNFRSF1B in RA synovium.** As one of the most abundant cell types in the synovium of RA, macrophages constitute the main source of the proinflammatory cytokine, TNF[22,23]. Thus, we obtained hyperplastic synovium from 5 RA patients to further analyze the interaction between fibroblasts and macrophage TNF signaling. We cataloged cells into 10 distinct cell lineages annotated with the expression levels of canonical marker genes (Fig. 4a). The fibroblasts mainly comprised four distinct subpopulations (Fig. 4b). To further characterize the functions of fibroblast clusters, we compared pathway activities and subdivided the fibroblast clusters into two subsets according to the observed significant phenotypic diversity (Fig. 4c, d). TNF signaling through NF-κB was enriched in fibroblast Clusters 1 and 2 (named Fib a) compared to fibroblast Clusters 3 and 4 (named Fib b) (Fig. 4c). Gene-expression network analysis further highlighted TNF signaling as the upregulated genes functional category in Fib a (Fig. 4e). Moreover, Fib a showed higher expression levels of ferroptosis-resistant synovial fibroblast marker genes in arthritis model mice (Fig. 4f). Functional analyses revealed that Fib b expressed genes involved in protein digestion and absorption and extracellular matrix-receptor interaction (Fig. 4g), which is consistent with the ferroptosis-sensitive subset in arthritis model mice (Fig. 3h). To delineate the molecular associations underlying the relationships between macrophages and the fibroblast subsets in RA patients, we constructed a cellular communication network using potential receptor-ligand pair interactions. In the network, interactions between macrophages and Fib a were more significant than those between macrophages and Fib b (Fig. 4h). Furthermore, we predicted the molecular interactions involved in the signaling of crucial cytokines in RA and found that fibroblasts transmitted most of the TNF signals via TNFRSF1A and TNFRSF1B (also known as TNFR1 and TNFR2) (Fig. 4i). In addition, TNF provided by macrophages showed a higher interaction with TNFR2 in Fib a than in the Fib b cluster (Fig. 4i). To confirm the involvement of TNFR2 in the regulation of NF-κB activity in RA synovial fibroblasts, we treated fibroblasts with anti-TNFR1 and anti-TNFR2 antibodies under long-term stimulation with TNF. We found that blocking TNFR2 abrogated NF-κB activation compared to blocking TNFR1, as measured by IκB phosphorylation (Fig. 4j). Thus, the stronger interaction between TNF from macrophages and TNFR2 in the Fib a cluster is consistent with the enrichment of TNF signaling through NF-κB in these

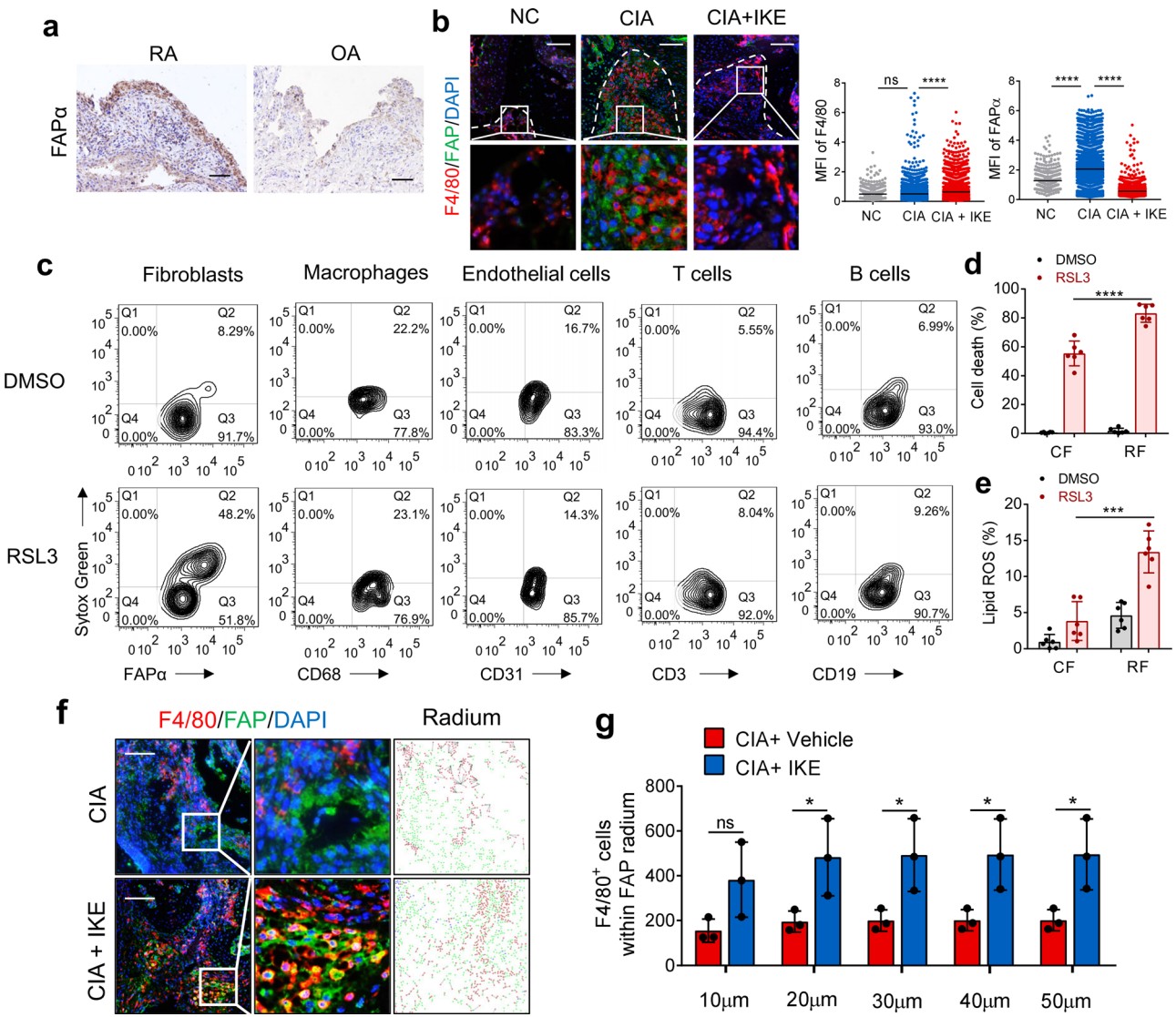

**Fig. 2 The ferroptosis inducer IKE decreased fibroblast populations in RA synovium. a** Representative immunohistochemical staining of FAPα in hyperplastic rheumatoid synovium of 26 RA and 21 OA patients. Scale bars, 50 μm. **b** Representative fluorescent multiplex IHC staining and scoring of joints labeled with anti-F4/80 (red), anti-FAPα (green), and DAPI (blue). Scale bars, 100 μm. $n = 413$ (NC), 3698 (CIA), 3469 (CIA + IKE) cells from 3 independent joints for each group. ****$P < 0.0001$; ns, $P = 0.6849$; one-way ANOVA followed by multiple comparisons. **c** Cell death in FAPα+ fibroblasts, CD68+ macrophages, CD31+ endothelial cells, CD3+ T cells, or CD19+ B cells isolated from hyperplastic synovium treated with 0.125 μM RSL3 for 6 h, quantified by SYTOX staining followed by flow cytometry. Cell death (**d**) and lipid ROS production (**e**) in circulating fibrocytes from PBMCs and synovial fibroblasts from inflamed joint fluid of RA patients treated with 0.125 μM RSL3 for 18 h (cell death) or 6 h (lipid ROS). ****$P < 0.0001$, ***$P = 0.0001$; two-tailed $t$-test. $n = 6$ patients. **f** Representative fluorescent multiplex IHC staining of retained hyperplasia synovium of CIA model mice treated with or without IKE and labeled with anti-F4/80 antibody (red), anti-FAPα antibody (green), and DAPI (blue). Scale bars, 200 μm. **g** The number of F4/80+ macrophages within the 50 μm radium of FAPα+ fibroblasts in joints of CIA mice. *$P < 0.05$; ns, $P = 0.00878$; two-tailed $t$-test. $n = 3$ joints. Data are presented as mean ± SD. Source data are provided as a Source data file.

fibroblasts. Altogether, these data suggest that the existence of fibroblast subsets with distinct ferroptosis sensitivity is probably related to the transduction of TNF signaling from macrophages.

**TNF protects RA synovial fibroblasts from ferroptosis while IL-6 and TGF-β sensitize fibroblasts to ferroptosis.** In the inflammatory cascade of RA, TNF is one of the key proinflammatory cytokines that enhances the activation of fibroblasts[24]. In turn, activated fibroblasts further enhance the inflammatory cycle and destruction of cartilage and bone by producing matrix metalloproteinases, chemokines, and inflammatory cytokines such as IL-6 and IL-8[25]. To identify factors that modulate susceptibility to ferroptosis, we primed human synovial fibroblasts from patients with RA with TNF and IL-6, which are

excessively produced inflammatory cytokines in RA, as well as with TGF-β, which has been well documented to be involved in fibrosis development[26]. As shown in Supplementary Fig. 8a and Fig. 5a–d, we found that in human synovial fibroblasts, TNF conferred significant and dose-dependent resistance to ferroptosis, and the associated lipid peroxidation induced by both IKE and RSL3 (IC₅₀ of IKE increased from 0.65to 1.16 μM, and IC₅₀ of RSL3 increased from 0.042 to 0.252 μM at 12 h).

Interestingly, TNF could protect synovial fibroblasts from only low-dose RSL3, while high doses of RSL3 still induced potent ferroptosis in the presence of TNF (Fig. 5b). On the other hand, both IL-6 and TGF-β effectively sensitized fibroblasts to RSL3- and IKE-induced cell death and lipid ROS (Fig. 5a–d and Supplementary Fig. 8b, c), and the effects were abolished by ferrostatin-1, an inhibitor

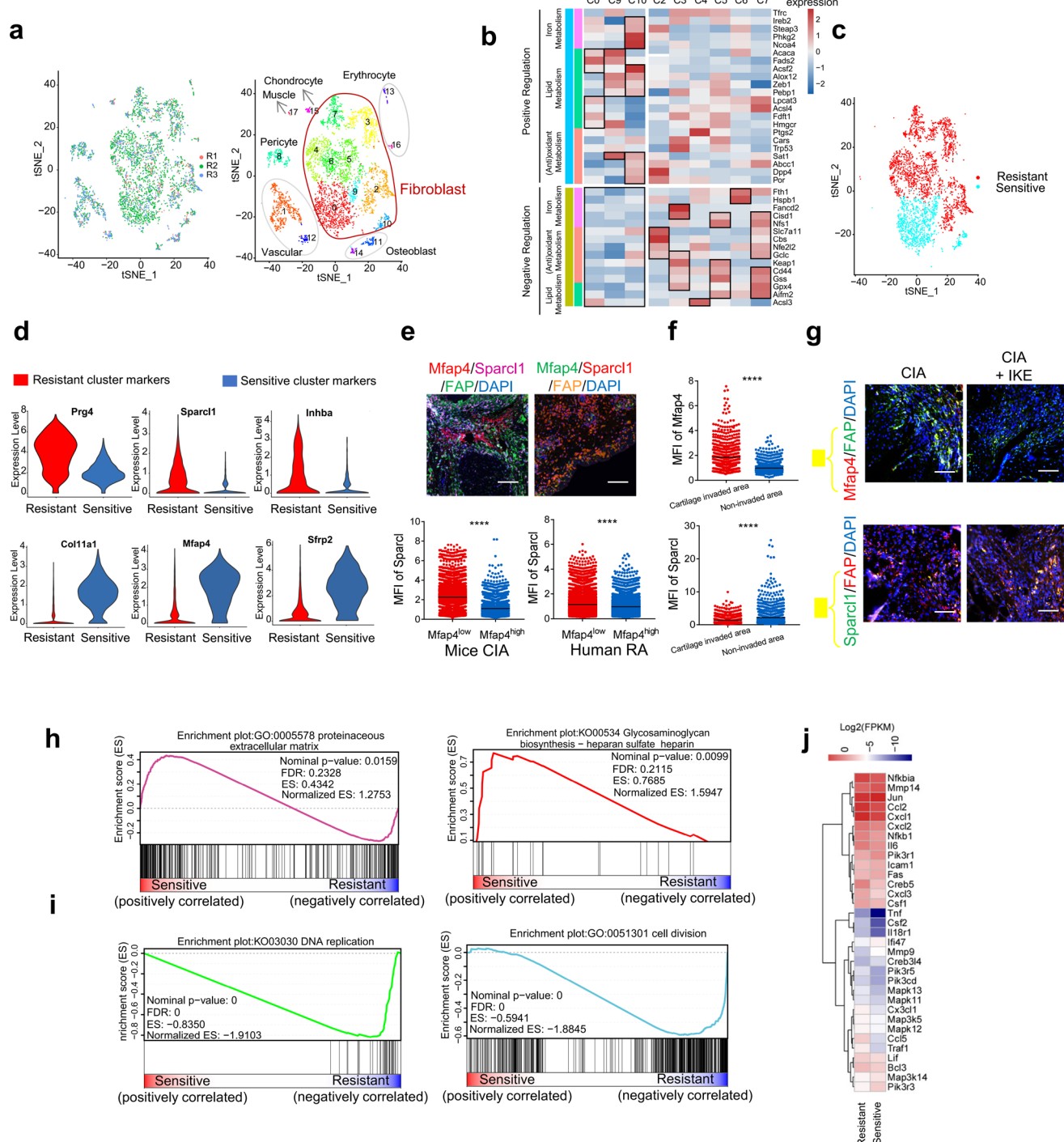

of ferroptosis (Supplementary Fig. 8d, e). The decrease in ferroptosis and lipid ROS levels induced by TNF could be eliminated by adalimumab, a fully human monoclonal TNF antibody (Fig. 5e, f) that has been approved for the treatment of RA, juvenile idiopathic arthritis, ankylosing spondylitis, and inflammatory bowel disease[27]. Tocilizumab, a humanized anti-IL-6 receptor monoclonal antibody[28], inhibited IL-6-enhanced cell death and lipid peroxidation in fibroblasts treated with IKE or RSL3 (Fig. 5g). The increased ferroptotic response caused by TGF-β was also effectively rescued by SB431542, a selective inhibitor of activin receptor-like kinase (ALK) receptors that are type I receptors in the TGF-β superfamily (Fig. 5h). One possible explanation for the protective effect of TNF is that TNF enhances cell proliferation and that higher cell confluence confers resistance to ferroptosis through cell–cell contacts[29]. However, in TNF-primed fibroblasts treated with IKE or RSL3, reduced cell density did not completely inhibit the defense against ferroptosis induced by TNF (Supplementary Fig. 9a–d). In contrast, the increased cell toxicities to ferroptosis inducers mediated via IL-6 and TGF-β were partially abolished by the higher cell confluence (Supplementary Fig. 9e–l).

**TNF promotes cystine uptake and the biosynthesis of cellular GSH through NF-κB signaling to protect synovial fibroblasts from lipid peroxidation and ferroptosis insult.** To investigate the molecular mechanism underlying the protective effect of TNF

**Fig. 3 Single-cell RNA sequencing reveals fibroblast subsets with distinct ferroptosis sensitivity. a** tSNE plot displaying 4735 CD45− cells from 3 STIA mice separated into 7 major cell types, including fibroblasts (9 clusters), vasculature (2 clusters), pericytes (1 cluster), chondrocytes (1 cluster), muscle cells (1 cluster), osteoblasts (2 clusters), and erythrocytes (2 clusters). **b** Heatmap showing the expression of high-risk ferroptosis genes in fibroblast clusters in the inflamed STIA joints. **c** Predicted ferroptosis-sensitive and ferroptosis-resistant fibroblast subsets based on ferroptosis-related genes listed in (**b**). **d** Expression of identified conserved marker genes in the ferroptosis-sensitive and ferroptosis-resistant fibroblast subsets. **e** Left, representative fluorescent multiplex IHC staining and quantification of inflamed joints of CIA mice labeled with anti-Mfap4 (red), anti-Sparcl1 (purple), anti-FAPα (green), and DAPI (blue). Mfap4$^{low}$, $n = 9905$ cells from 10 inflamed joints. Mfap4$^{high}$, $n = 3336$ cells from 10 inflamed joints. Right, representative fluorescent multiplex IHC staining (top panel) and quantification (bottom panel) of hyperplastic synovium of RA patients labeled with anti-Mfap4 (green), anti-Sparcl1 (red), anti-FAPα (orange), and DAPI (blue). Mfap4$^{low}$, $n = 8222$ cells from 10 independent human synovium samples. Mfap4$^{high}$, $n = 6730$ cells from 10 independent human synovium samples. Scale bars, 200 μm. ****$P < 0.0001$; two-tailed $t$-test. **f** Quantification of the mean fluorescence intensity (MFI) of Mfap4 (top) and Sparcl1 (bottom) in hyperplastic tissue invading cartilage and in the remaining tissue. ****$P < 0.0001$; two-tailed $t$-test. Cartilage invaded area, $n = 684$ cells from 3 inflamed joints of CIA mice. Non-invaded area, $n = 1257$ cells from 3 inflamed joints of CIA mice. **g** Representative immunofluorescent staining of inflamed joints of CIA model mice with or without treatment with a low dose of IKE and labeled with anti-Mfap4, anti-Sparcl1, anti-FAPα, and DAPI. Similar results were observed from 15 joints tested for each group. Scale bars, 100 μm. **h** Enrichment plots of the GO_Proteinaceous extracellular matrix gene set (left) and KEGG_Glycosaminoglycan biosynthesis gene set (right) among the ferroptosis-sensitive subsets identified by GSEA. The $p$-value is calculated through permutation tests. **i** Enrichment plots of the GO_Cell division gene set (left) and KEGG_DNA replication gene set (right) among the ferroptosis-resistant subset identified by GSEA. The $p$-value is calculated through permutation tests. **j** Expression of selected mRNA transcripts in the TNF pathway gene set among ferroptosis-sensitive and ferroptosis-resistant subsets. Source data are provided as a Source data file.

against ferroptosis, we tested intracellular iron storage and GSH levels. Iron is actively involved in the initiation of lipid peroxidation via the Fenton reaction and its incorporation into certain enzymes, such as LOX and POR[6]. The cystine/cysteine/GSH/GPX4 axis has been recognized as one of the main pathways that protects cells from ferroptosis[6]. The exhaustion of GSH could directly impact GPX4 activity and stability, leading to ferroptosis[30]. We found that IKE resulted in the depletion of GSH, whereas TNF increased the level of GSH and mitigated IKE-driven exhaustion of GSH in synovial fibroblasts (Fig. 6a). Meanwhile, TNF did not affect the availability of labile iron (Supplementary Fig. 10a). To explore the underlying mechanism responsible for TNF-promoted GSH production, we tested crucial molecules for GSH biosynthesis. As a key component of the sodium-independent cystine-glutamate antiporter, system xc (xCT), SLC7A11 is responsible for the generation of intracellular GSH[31]. Glutamate-cysteine ligase, the first rate-limiting enzyme of GSH biosynthesis, consists of glutamate-cysteine ligase regulatory subunit (GCLC) and glutamate-cysteine ligase catalytic subunit (GCLM)[32]. We found that the levels of *SLC7A11*, *GCLC*, and *GCLM* were significantly increased upon TNF treatment (Fig. 6b), suggesting that these genes might be transcriptionally regulated. Effective knockdown of *SLC7A11* sensitized TNF-treated fibroblasts to ferroptosis upon RSL3 treatment (Fig. 6c, d). Furthermore, the GCLC inhibitor buthionine sulfoximine (BSO) negated TNF-induced ferroptosis resistance (Fig. 6e), and knockdown of *GCLC* sensitized TNF-treated fibroblasts to ferroptosis (Fig. 6f, g). These results indicated that TNF confers robust protection against ferroptosis mainly by enhancing biosynthesis and restoring GSH levels. NF-κB and mitogen-activated protein (MAP) kinase/ activator protein 1 (AP-1) (c-Fos/c-Jun) are two pivotal proteins downstream of TNF that are activated in synovial fibroblasts[33]. Notably, knockdown of NF-κB p65 in fibroblasts resulted in markedly reduced cell viability and increased cell death in the presence of RSL3 and TNF (Fig. 6h, i) in tandem with the inhibition of TNF-induced upregulation of *SLC7A11*, *GCLC*, and *GCLM* expression (Fig. 6j). PS1145 is an inhibitor of IκB kinase (IKK) that specifically inhibits IKK-mediated IκB phosphorylation; otherwise, NF-κB is released from its complex with IκB and subsequently translocates to the nucleus[34]. We then found that PS1145 effectively inhibited NF-κB activation and abrogated the upregulation of *SLC7A11*, *GCLC*, and *GCLM* expression (Fig. 6k, l). In addition, PS1145 obviously hindered TNF-mediated defense against ferroptosis and lipid ROS accumulation (Fig. 6m, n, Supplementary Fig. 10b).

Consistently, TNF-induced GSH accumulation was effectively terminated upon PS1145 treatment (Fig. 6o). Activated fibroblasts from individuals with RA exhibit similar proliferative features as tumor cells and are contact-inhibited at high densities in culture. Direct cell–cell interactions between these fibroblasts seem to be an important feature for fibroblast function[35,36]. To better mimic an in vivo microenvironment, we cultured fibroblasts into 3D multicellular spheroids. Consistent with the results from the 2D analysis, the combination of PS1145 and RSL3 partially ameliorated TNF-induced protection against ferroptosis and triggered cell death in fibroblast spheroids (Supplementary Fig. 10c). These results indicate that NF-κB signaling is responsible for the protective role of TNF against ferroptosis by maintaining GSH biosynthesis. Although AP-1 was also found to enhance the expression of GCLC[37], knockdown of the canonical AP-1 transcription factor c-Jun failed to inhibit the protection conferred by TNF against ferroptosis (Supplementary Fig. 10d). It has been reported that TNF could promote the activation of NADPH oxidase (NOX) and intracellular ROS generation by the NF-κB pathway[38]. Since ferroptosis can be induced by the accumulation of superoxide and hydrogen peroxide upon upregulation of NOX, we wondered whether TNF/NFκB may also activate ferroptosis-sensitizing signaling. Here, we showed that short-term exposure of fibroblasts to TNF markedly increased intracellular ROS levels, which were reduced by pretreatment with an inhibitor of NOX (NOXi) (Fig. 6p). However, long-term exposure of fibroblasts to TNF failed to result in permanent increase in ROS (Fig. 6p). Of note, NOXi further enhanced the protective effect of TNF against ferroptosis triggered by RSL3 (Fig. 6q). These data suggest that the defensive effect of TNF against ferroptosis overcomes the contribution of TNF/NFκB-induced ROS generation. We also found that sublethal concentrations of RSL3 resulted in markedly reduced invasion and migration capacities and decreased MMP8, MMP9, MMP11, MMP13, and MMP14 expression in TNF-induced fibroblasts with NF-κB p65 subunit knockdown (Supplementary Fig. 10e–h). These results indicated a critical contribution of sublethal doses of ferroptosis inducers to the modulation of the aggressive behaviors of RA synovial fibroblasts.

**IL-6 sensitizes synovial fibroblasts to ferroptosis by increasing cellular free iron contents but has no effect on cellular GSH.** In contrast to TNF, IL-6, another pleiotropic cytokine secreted in large amounts by activated fibroblasts in the pathogenesis of RA, increased the toxicities of ferroptosis inducers. Therefore, we investigated the molecular mechanism by which IL-6 sensitizes

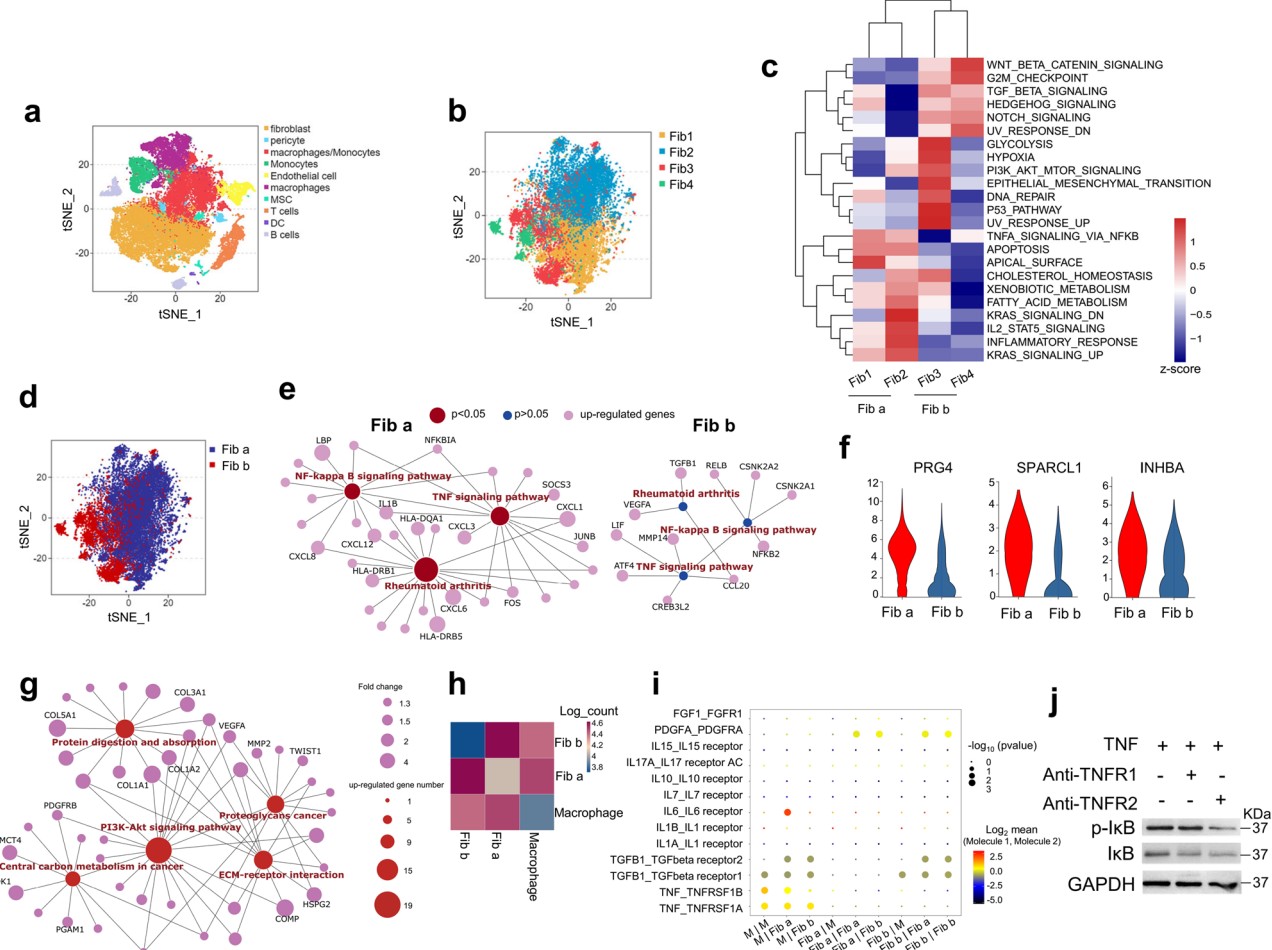

**Fig. 4 Single-cell RNA sequencing reveals the crosstalk between macrophages and the fibroblast subsets through TNF signal. a** Top, tSNE plot displaying 56396 cells from the synovium of 5 RA patients separated into 10 major cell types, including fibroblasts, pericytes, macrophages, monocytes, endothelial cells, MSCs, T cells, DCs, and B cells. **b** identification of 4 distinct fibroblast clusters. **c** Differences in pathway activities scored by GSVA among the different fibroblast clusters. **d** Identification of two fibroblast subsets according to the pathway activities in (**c**). **e** Functional association networks between signature genes specific to Fib a and Fib b subsets. **f** Violin plots showing expression levels for ferroptosis-resistant fibroblast marker genes in the identified Fib a and Fib b subsets. **g** Functional association networks between signature genes specific to Fib b subset. **h** Heat map showing the relative interaction between macrophages and the fibroblast subsets. **i** Overview of selected ligand–receptor interactions. The p-value is calculated through permutation tests. P values are indicated by circle size. The means of the average level of interacting molecule 1 in cluster 1 and interacting molecule 2 in cluster 2 are indicated by color. M, macrophages. **j** Western blotting analysis of p-IκB and IκB expression in fibroblasts treated with TNF and the specific blocking antibodies for TNFR1 or TNFR2 for 48 h. TNFR1 neutralizing antibody (Sino Biological, 10872-R111), TNFR2 neutralizing antibody (Sino Biological, 10417-R00N6). Source data are provided as a Source data file.

fibroblasts to ferroptosis and found that IL-6 alone led to an increase in the cellular labile iron pool (LIP) level without affecting the GSH level (Supplementary Fig. 11a, b). In addition, IL-6 decreased the expression of SLC40A1 and ferritin (Supplementary Fig. 11c). Further analysis revealed that knockdown of SLC40A1 markedly increased the toxicity of RSL3-driven ferroptosis (Supplementary Fig. 11d–f). Tumor cells often show a marked alteration in metabolism leading to the intracellular accumulation of iron, which is heavily utilized for tumor growth and angiogenesis[39]. Here, IL-6 effectively induced an increase in the intracellular iron level, which may also contribute to the proliferation of RA synovial fibroblasts.

**Combination of TNF blockade with ferroptosis induction leads to fibroblast ferroptosis and reduced cartilage and bone damage in vivo.** Owing to the role of TNF in protecting synovial fibroblasts of RA patients from ferroptosis, we sought to determine the effects of a low dose of IKE in combination with TNF

blockade in the CIA mouse model. Here, we used etanercept, an anti-TNF antagonist that has been approved for the treatment of inflammatory diseases (including RA) and has been reported to effectively block mouse TNF[40]. The combination of etanercept and IKE undermined TNF-induced protection against ferroptosis and triggered cell death in spheroids formed by synovial fibroblasts (Fig. 7a). Etanercept also markedly increased the sensitivity of fibroblasts from individuals with RA primed with fibroblast supernatant to ferroptosis (Fig. 7b), suggesting that TNF secreted by fibroblasts may also contribute to the generation of an autocrine ferroptosis prevention network. After active inflammation developed, the CIA mice were treated with a low dose of etanercept (2 mg/kg, twice/week) and/or a reduced dose of IKE (20 mg/kg, twice/week) for 5 weeks. While a low dose of either IKE or etanercept alone showed little effect on the severity of joint swelling and bone and cartilage destruction, etanercept in combination with IKE attenuated the severity of synovial inflammation and blocked disease development (Fig. 7c–e). The combination therapy regimen also led to reduced cartilage and

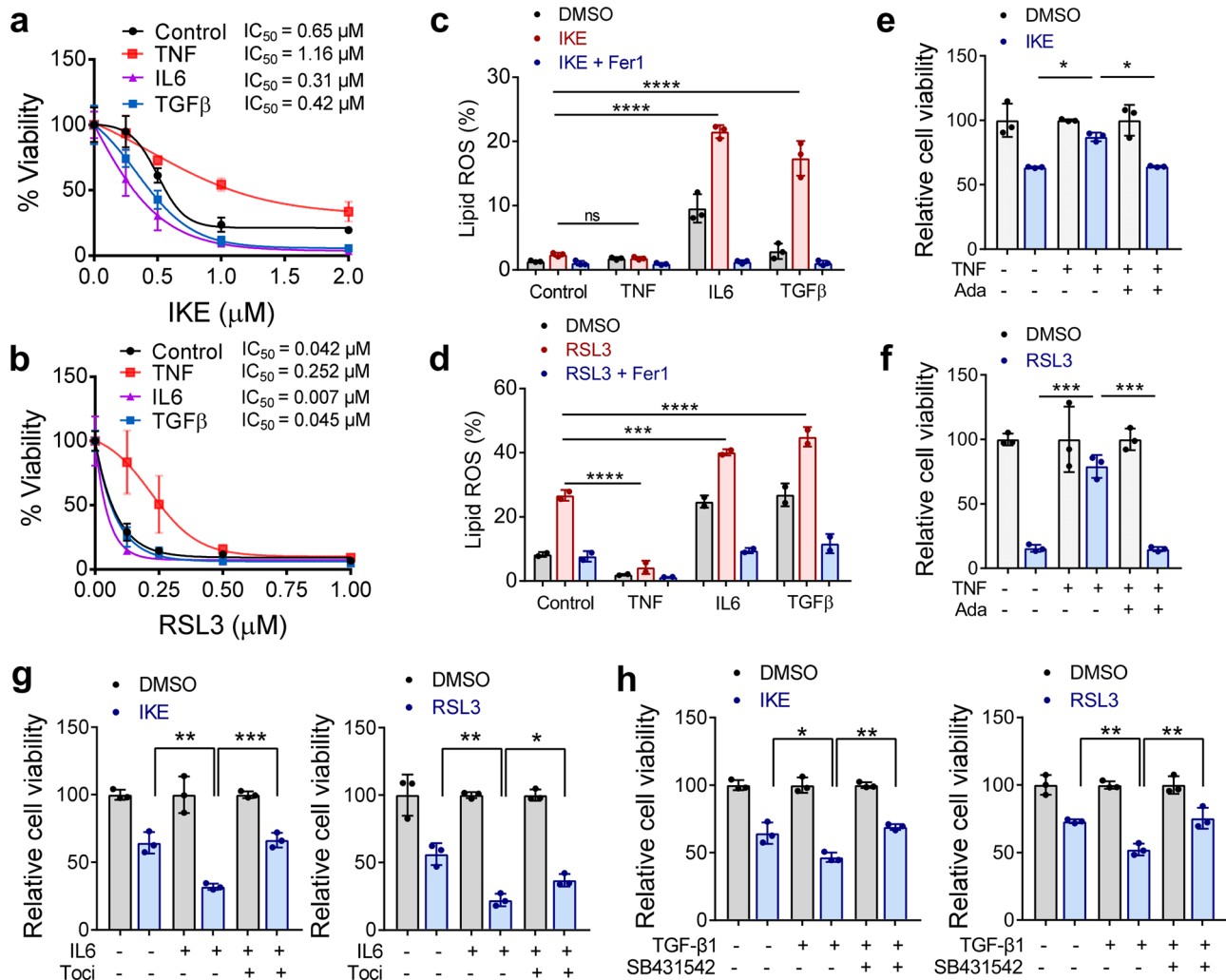

**Fig. 5 TNF protects RA fibroblasts from ferroptosis while IL-6 and TGF-β sensitize fibroblasts to ferroptosis. a, b** Relative viability of fibroblasts derived from RA patients and primed with TNF, IL-6, or TGF-β (20 ng/ml) for 72 h, followed by treatment with different concentrations of IKE or RSL3. Cell viability was assayed by measuring cellular ATP levels at 26 h (IKE) (**a**) or 12 h (RSL3) (**b**) after treatment. $n = 3$ biologically independent samples per condition. IC$_{50}$ values were calculated using nonlinear regression analysis. **c, d** Cells were treated as indicated for 18 h (IKE, 1 μM) (**c**) or 4 h (RSL3, 0.125 μM) (**d**) in the presence of the ferroptosis inhibitor Ferrostatin-1 (Fer1) (1 μM) and lipid ROS accumulation was measured by BODIPY C11 staining coupled with flow cytometry. $n = 2$ biologically independent samples per condition. ns, $P > 0.9999$, ****$P < 0.0001$, ***$P = 0.0007$; one-way ANOVA followed by multiple comparisons. **e, f** The relative viability of fibroblasts primed with TNF in the presence of the anti-TNF blocking antibody adalimumab (Ada) for 72 h, followed by treatment with IKE (**e**) or RSL3 (**f**). $n = 3$ biologically independent samples per condition. *$P = 0.0183$, 0.0216 (left to right), ***$P = 0.0003$, 0.0002 (left to right); one-way ANOVA followed by multiple comparisons. **g** Relative viability of fibroblasts primed with IL-6 in the presence of the anti-IL-6 receptor monoclonal antibody tocilizumab (Toci), followed by treatment with IKE or RSL3. $n = 3$ biologically independent samples per condition. Left, **$P = 0.0011$, ***$P = 0.0008$; right, **$P = 0.0011$, *$P = 0.0182$; one-way ANOVA. **h** Relative viability of fibroblasts primed with TGF-β in the presence of the ALK receptor inhibitor SB431542 followed by treatment with IKE or RSL3. $n = 3$ biologically independent samples per condition. Left, *$P = 0.0131$, **$P = 0.0042$; right, **$P = 0.0067$, **$P = 0.0040$; one-way ANOVA followed by multiple comparisons. All bar graphs are represented as mean ± SD. Source data are provided as a Source data file.

bone damage (Fig. 7f–h) and markedly decreased the population of FAPα+ fibroblasts, confirming that TNF inhibition sensitized fibroblasts to ferroptosis induction (Fig. 7i). The levels of p-NF-κB, GCLM, and GCLC were significantly reduced in the hyperplastic synovium of CIA mice treated with etanercept, indicating the effective inhibition of TNF/NF-κB signaling (Fig. 7j). Immunohistochemical analysis of PTGS2 and GPX4 expression supported the synergistic effect of the combined inhibition of TNF signaling and cystine uptake on inducing fibroblast ferroptosis in vivo (Fig. 7k). These results confirm that cystine deprivation by IKE can synergize with the TNF blockade to mitigate fibroblast activity by inducing ferroptotic cell death. To evaluate

the risk of normal tissue damage likely caused by long-term ferroptosis induction, we collected and stained the main organs, including hearts, livers, spleen, lung, kidney, and knee joints of IKE-treated mice. No obvious pathological changes in the main organs of CIA mice treated with low dose IKE were observed (Supplementary Fig. 12a).

## Discussion

Numerous studies have shown that ferroptosis is involved in certain disease contexts, such as ischemia/reperfusion injury, neural degenerative diseases, and tumors[6]. Recently, ferroptosis has been shown to involve in cell death of neutrophils and

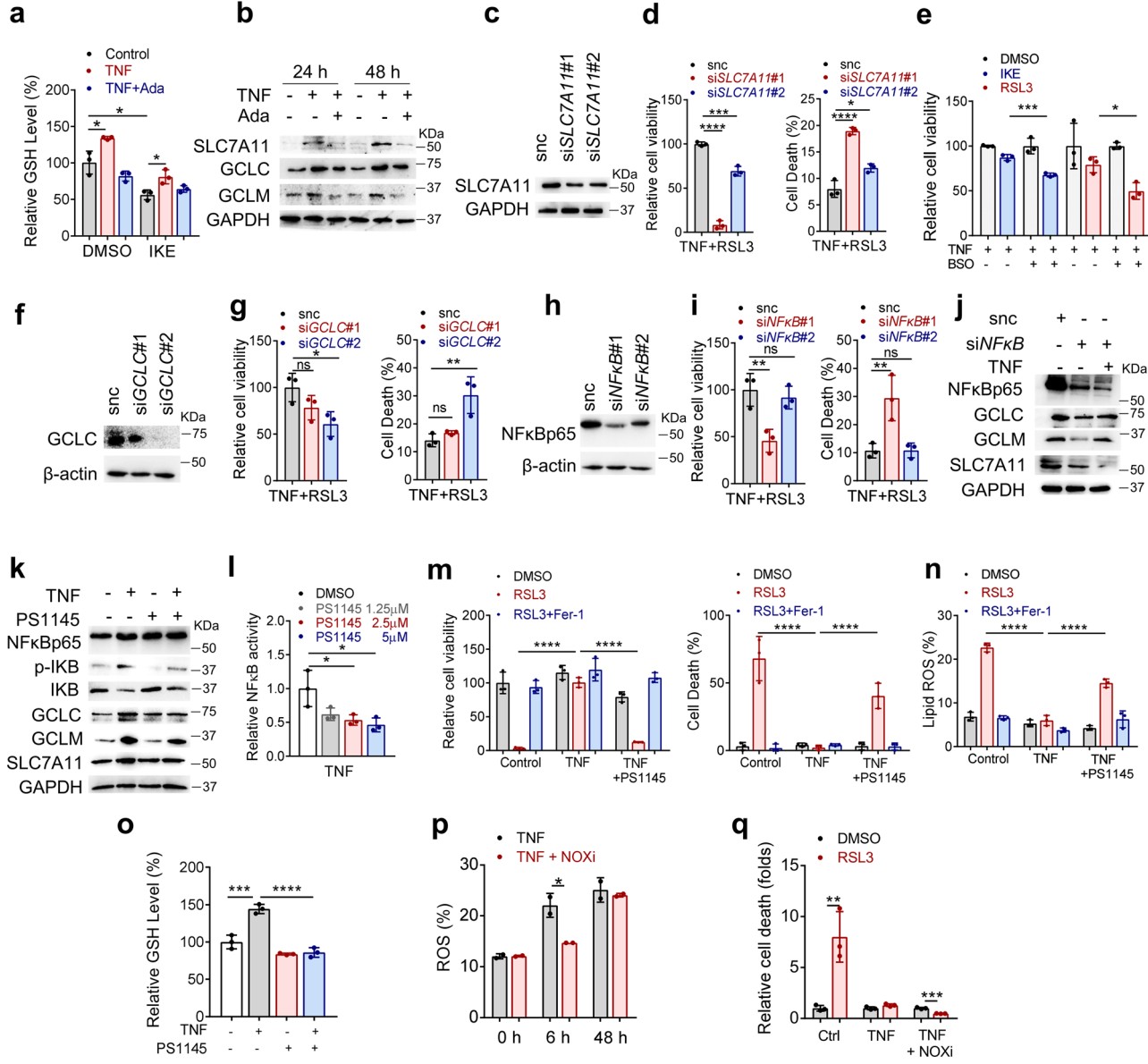

follicular helper T (Tfh) cells, which related to immune disorders[41,42]. Our data showed that synovial fibroblasts from RA patients have increased iron accumulation and lipid peroxidation. We found that stress not only manifests from the already established surrounding microenvironment in the lining and sub-lining layers of RA synovium and joint effusion but also is enhanced by the continued release of various cytokines, including IL-6 and TGF-β. Oxidative stress and the accumulation of lipid peroxidation are important drivers for chronic inflammation in the joints[43]. Thus, modulation of the oxidative microenvironment is believed to be a promising strategy to improve the efficacy of RA treatment[18]. Interestingly, we found that the administration of a lipid ROS inhibitor could not suppress the development of inflammation and joint damage when started after the appearance of significant arthritis symptoms. High levels of lipid peroxidation and 4-HNE expression correlate with low in vivo oxygen tension levels, and animal studies show that hypoxia occurs at the pre-arthritic stage of RA[44]. In addition, the modification of proteins by cytotoxic 4-HNE may contribute to the onset of autoimmune reactions or even autoimmune disease processes[45]. Taken together, our data indicated that lipid peroxidation considerably contributes to the onset of RA pathogenesis.

TNF, known to be a master driver of RA progression, stimulates the proliferation of synovial fibroblasts and initiates a chronic synovitis that promotes cartilage destruction and bone erosion[24]. Here we found that TNF can inhibit ferroptosis independent of its proinflammatory function. TNF enhanced cystine uptake and cellular GSH biosynthesis via activation of the NF-κB pathway to protect fibroblasts from oxidative stress and excessive iron. Interestingly, it has been reported that TNF/TNFR1/NF-κB signaling could promote the activation of NOX[38]. NOX is responsible for the generation of ROS and may contribute to the sensitization of ferroptosis[46]. Indeed, our data showed that ROS are induced when fibroblasts are treated with TNF in the short-term. However, long-term exposure to TNF failed to further induce ROS, and inhibition of NOX enhanced the protective effect of TNF against ferroptosis. Although both TNFR1 and TNFR2 signaling have been proven to initiate NF-κB function via activation of TRAF-2, the increase in NF-κB activity in response to TNFR1 activation is rapid and transient, whereas TNFR2 activation results in a much slower but persistent response[47]. Consistently, we found that long-term stimulation with TNF tended to activate NF-κB signaling through TNFR2 but not TNFR1. These data suggest that the defensive effect of TNF/TNFR2/NF-κB against ferroptosis overcomes the

**Fig. 6 TNF protects RA fibroblasts from the insult of lipid peroxidation and ferroptosis by enhancing the GSH biosynthetic pathway via activation of NF-κB. a** Intracellular GSH in fibroblasts treated with TNF in the presence of adalimumab followed by IKE treatment for 18 h. $n = 3$ biologically independent samples per condition. *$P = 0.0211$, 0.0224, 0.0109 (left to right); two-tailed $t$-test. **b** Western blotting analysis of GCLC, GCLM, and SLC7A11 expression in fibroblasts at 24 h and 72 h after exposure to TNF and adalimumab. **c** Western blotting analysis of SLC7A11 in fibroblasts transfected with scramble siRNA or either of two independent siRNAs targeting SLC7A11 (siSLC7A11#1, siSLC7A11#2). **d** Relative cell viability and cell death of fibroblasts transfected with scramble siRNA or siRNA targeting SLC7A11, primed with TNF and then treated with RSL3. $n = 3$ biologically independent samples per condition. ****$P < 0.0001$, ***$P = 0.0003$, *$P = 0.0101$; one-way ANOVA followed by multiple comparisons. **e** Relative viability of fibroblasts from individuals with RA and primed with TNF in the presence of the GCLC inhibitor BSO, followed by treatment with IKE (1 μM) or RSL3 (0.125 μM). ***$P = 0.0001$, *$P = 0.0167$; two-tailed $t$ test. **f** Western blotting analysis of GCLC expression in fibroblasts transfected with scramble siRNA or either of two independent siRNAs targeting GCLC (siGCLC#1, siGCLC#2). **g** Relative viability and cell death of fibroblasts transfected with scramble siRNA or siRNA targeting GCLC, primed with TNF, and then treated with RSL3. Cell viability was assayed at 12 h, and cell death was measured at 10 h. $n = 3$ biologically independent samples per condition. Left, ns, $P = 0.2174$, *$P = 0.0314$; right, ns, $P = 0.7042$, **$P = 0.0064$; one-way ANOVA followed by multiple comparisons. **h** Western blotting analysis of NF-κB p65 expression in fibroblasts transfected with scramble siRNA or either of two independent siRNAs targeting NF-κB p65 (siNF-κB#1, siNF-κB#2). **i** Relative viability and cell death of fibroblasts transfected with scramble siRNA or siRNA targeting NF-κB p65, primed with TNF and then treated with RSL3. $n = 3$ biologically independent samples per condition. Left, ns, $P = 0.7675$, **$P = 0.0079$; right, ns, $P = 0.9997$, **$P = 0.0099$; one-way ANOVA followed by multiple comparisons. **j** Western blotting analysis of NF-κB p65, GCLC, GCLM, and SLC7A11 expression in fibroblasts transfected with scramble siRNA or siRNA targeting NF-κB p65 and treated with TNF for 72 h. **k** Western blotting analysis of NF-κB p65, p-IκB, IκB, GCLC, and GCLM expression in fibroblasts treated with TNF and the IκB kinase inhibitor PS1145 for 72 h. **l** Relative NF-κB activity in fibroblasts treated with TNF and PS1145. $n = 3$ biologically independent samples per condition. *$P = 0.0263$, 0.0123 (left to right); one-way ANOVA followed by multiple comparisons. **m** Relative viability and cell death in fibroblasts primed with TNF and PS1145 and then treated with RSL3 and ferrostatin-1. $n = 3$ biologically independent samples per condition. ****$P < 0.0001$; one-way ANOVA followed by multiple comparisons. **n** Lipid ROS levels in fibroblasts primed with TNF and PS1145 and then treated with RSL3 and ferrostatin-1. $n = 3$ biologically independent samples per condition. ****$P < 0.0001$; one-way ANOVA followed by multiple comparisons. **o** Intracellular GSH in fibroblasts treated with TNF in the presence of PS1145. $n = 3$ biologically independent samples per condition. ***$P = 0.0001$, ****$P < 0.0001$; one-way ANOVA followed by multiple comparisons. **p** ROS levels in fibroblasts primed with TNF and NOX inhibitor (NOXi) for 6 or 48 h. $n = 2$ biologically independent samples per condition. *$P = 0.0469$; two-tailed $t$ test. **q** Relative cell death in fibroblasts primed with TNF and NOXi for 48 h and then treated with RSL3. $n = 3$ biologically independent samples per condition. **$P = 0.0083$, ***$P = 0.0003$; two-tailed $t$ test. All bar graphs are presented as mean ± SD. Source data are provided as a Source data file.

ferroptosis-sensitizing contribution of TNF/TNFR1/NFκB-induced ROS production.

Activation of multiple pathways upon oxidative stress means that basal TNF stimulation in cells that express more ferroptosis-sensitive genes allows the accumulation of lipid peroxides, whereas potent and sustained TNF stimulation in cells with more ferroptosis-resistant genes would strongly prevent lipid peroxide accumulation and ferroptosis induction. The single-cell transcriptome of murine synovial fibroblasts showed that ferroptosis-resistant fibroblasts expressed proliferation-related genes that regulate the cell cycle, whereas ferroptosis-sensitive fibroblasts expressed genes related to extracellular matrix formation. Consistently, an in vitro assay showed that long-term exposure of fibroblasts to TNF increased the genes related to proliferation and decreased the genes related to extracellular matrix, implying the potential regulatory role of long-term stimulation of TNF and lipid peroxidation stress in diversification of fibroblast functions.

Recent studies suggest that pharmacological modulation of lipid peroxidation and ferroptosis elicits a significant clinical benefit for certain diseases[6]. Of note, ferroptosis induction is emerging as a potential approach for cancer treatment when used either alone or in combination with immunotherapy, radiotherapy, and some targeted therapies[6]. On the other hand, ferroptosis contributes to ischemic organ injury and neurodegeneration, thus a more pragmatic approach would be to consider modulating the ferroptotic pathway in a manner that renders the cells more susceptible to lower doses of ferroptosis inducers. We report that TNF protects synovial fibroblasts from oxidative stress and ferroptosis by enhancing cystine uptake and the cellular GSH biosynthetic pathway via upregulation of *SLC7A11*, *GCLC*, and *GCLM*. Although TNF blockade could not directly induce fibroblast ferroptosis, it sensitizes fibroblasts to ferroptosis inducers. The low-dose ferroptosis inducer, IKE, when combined with the anti-TNF antagonist etanercept, was able to trigger fibroblast ferroptosis and lead to reduced cartilage and bone damage in the CIA experimental arthritis model.

CIA model requires an intact adaptive immune response, particularly through the activation of pathogenic collagen-specific T cells and the generation of anti-collagen antibodies[48]. We found that the anti-type II collagen antibody in the serum of CIA mice did not differ significantly between mice treated with etanercept in combination with IKE and mice treated with the vehicle (Supplementary Fig. 12b), suggesting that the reduced inflammation and joint damage are not due to modulating autoantibody production. As a crucial cell type of the innate immune system, macrophages play a central role in initiating and driving the pathogenesis of RA. Compared with fibroblasts in healthy joints that have modest immune-regulatory functions, RA fibroblasts have emerged as important immune modulators that regulate the influx of the inflammatory infiltrate by secreting inflammatory factors and by engaging in crosstalk with neighboring immune cells, especially macrophages. In addition to the reaction to proinflammatory stimuli from macrophages, RA-related fibroblasts in the synovial lining contribute actively to the pathogenesis of RA by cooperating with macrophages through direct cell–cell interactions via ligation of CD55 on fibroblasts with CD97 on macrophages[49]. Prostaglandins secreted by RA fibroblasts work in collaboration with proinflammatory factors to shift macrophages toward a state characterized by high expression of pro-heparin-binding EGF-like growth factor (HBEGF), which further promotes fibroblasts invasiveness[50]. RANKL expressed by RA fibroblasts can also enhance osteoclastogenesis from macrophages[12]. Thus, ferroptosis induction seems to attenuate the interaction between fibroblasts and immune cells, which helps to mitigate inflammation and restore synovial homeostasis.

Although we did not observe obvious pathological changes in the main organs of CIA mice treated with low-dose IKE, long-term administration of ferroptosis inducers may increase the risk of lethal inflammation and tissue damage. Activated fibroblasts showed higher sensitivity to ferroptosis inducers compared with other main cell types in the hyperplastic synovium, which partially increased the specificity of ferroptosis induction. In addition, fibroblast-directed ferroptosis strategies should be

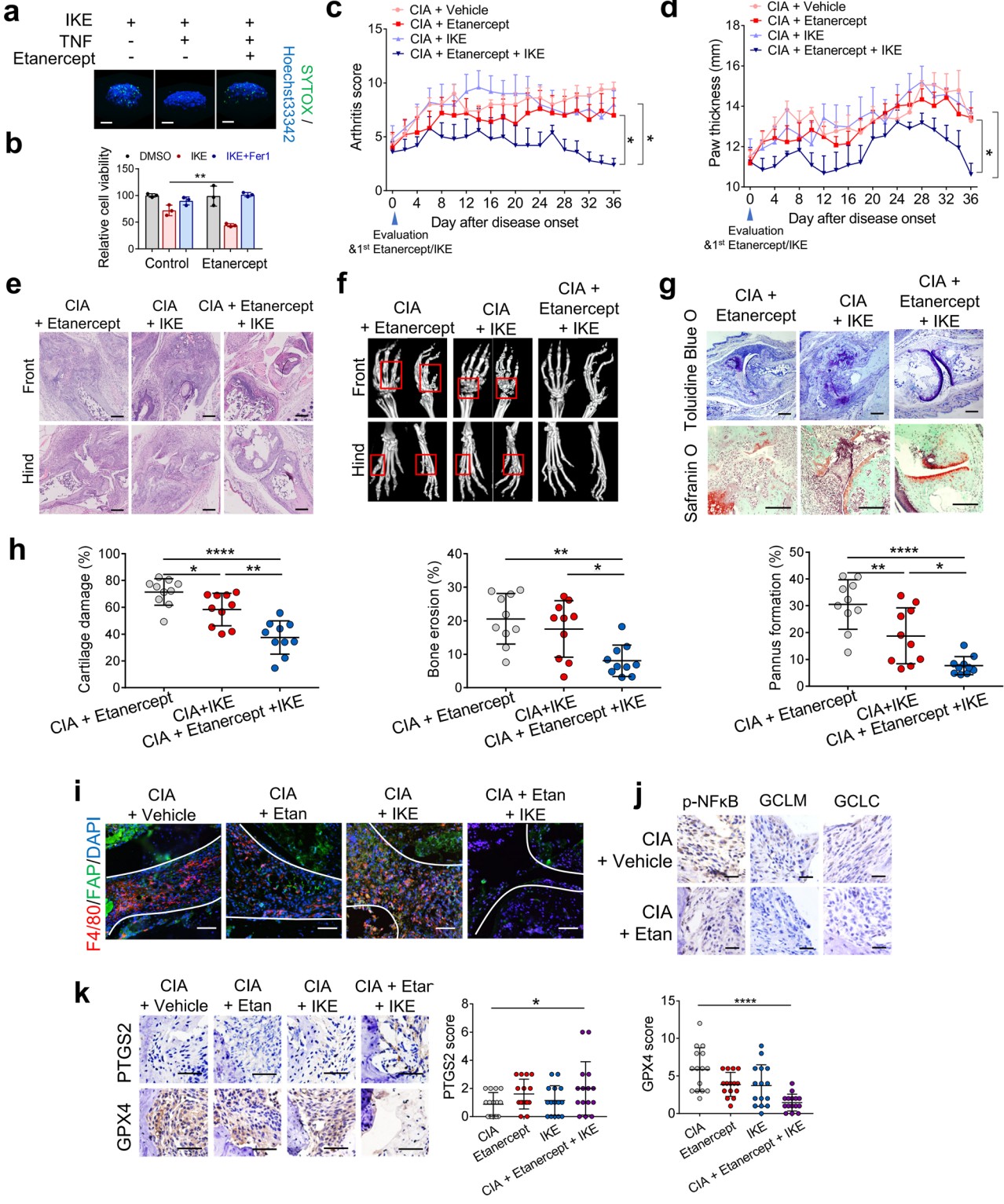

developed. Future studies will potentially identify surface proteins that are more specific to fibroblasts, which will facilitate ferroptosis strategies that target fibroblasts therapeutically. Moreover, an FDA-approved rheumatoid arthritis drug, sulfasalazine, was also proven to effectively trigger ferroptosis[51,52]. The combination of the drug together with anti-TNFα therapy could be tested to reduce the risk of ferroptosis induction of proto-drugs.

In conclusion, targeting ferroptosis-associated metabolism in RA has revealed exciting insights into the improvement of the efficacy of currently available therapeutic approaches. Furthermore, selectively targeting fibroblast subpopulations to induce ferroptotic cell death might be a promising therapeutic approach for a wide range of diseases associated with dysregulated fibroblast activation.

## Methods

**Human specimen research.** Human specimen research was approved by the Medical Ethics Committee of the First Affiliated Hospital (Xijing Hospital) of the

**Fig. 7 A TNF antagonist sensitizes RA fibroblasts to ferroptosis induction in CIA model mice. a** Three-dimensional spheroids formed by fibroblasts were primed with TNF and the TNF antagonist etanercept for 48 h before treatment with IKE (4 μM) for an additional 30 h. Dead cells were stained with SYTOX and Hoechst33342 for 1 h. Scale bars, 100 μm. **b** Relative viability of fibroblasts from patients with RA that were primed with fibroblast supernatant in the presence of the TNF antagonist etanercept for 48 h followed by treatment with IKE (1 μM) and ferrostatin-1 (Fer1, 1 μM) for an additional 26 h. $n = 3$ biologically independent samples per condition. $**P = 0.0092$; two-tailed $t$ test. **c, d** CIA model mice were intraperitoneally injected with 20 mg/kg IKE twice a week and/or 2 mg/kg etanercept twice a week for 5 weeks. Joint inflammation measured by arthritis score (**c**) ($*P = 0.0461$, CIA + Etanercept vs CIA + Etanercept+IKE; $**P = 0.0132$; CIA + IKE vs CIA + Etanercept+IKE; one-way ANOVA followed by multiple comparisons to compare the means at the end point) and paw thickness (**d**) ($*P = 0.0233$, CIA + Etanercept vs CIA + Etanercept+IKE; $*P = 0.0356$, CIA + IKE vs CIA + Etanercept+IKE; one-way ANOVA followed by multiple comparisons to compare the means at the end point). $n = 5$ mice for each group. **e** H&E staining images of representative joints in control and day 36 CIA model mice. Scale bars, 200 μm. **f** Representative micro-CT images of control and CIA mice treated with IKE and/or etanercept. **g** Toluidine blue O and safranin O staining images of representative joints. Scale bars, 200 μm. **h** Quantification of histomorphometric analysis of cartilage damage, bone erosion, and pannus formation. Left, $*P = 0.0447$, $**P = 0.0011$, $****P < 0.0001$; middle, $*P = 0.0150$, $**P = 0.0014$; right, $*P = 0.0152$, $**P = 0.0099$, $****P < 0.0001$; one-way ANOVA followed by multiple comparisons. $n = 10$ joints for each group. **i** Representative fluorescent multiplex IHC staining and quantification of joints labeled with anti-F4/80 (red), anti-FAPα (green), and DAPI (blue). Scale bars, 100 μm. Etan, Etanercept. **j** Immunohistochemical staining of p-NF-κB, GCLM, and GCLC in the joints of CIA model mice with or without etanercept treatment. Scale bars, 25 μm. **k** Immunohistochemical staining and scoring of PTGS2 and GPX4 expression in the joints of CIA mice. $n = 15$ joints for each group. $*P = 0.0425$, $****P < 0.0001$; one-way ANOVA followed by multiple comparisons. Scale bars, 50 μm. Data in **c** and **d** are presented as mean ± SEM. Other data are presented as mean ± SD. Source data are provided as a Source data file.

Fourth Military Medical University (KY20192006-F-1). Informed written consent was obtained from all patients/patients' families prior to participation. The biopsies were obtained from key-hole arthroscopy (during arthroscopic cleansing of the knee) and the synovial fluid samples were obtained during joint aspiration. No patients included in the study were being under corticosteroids treatment or second-line drug agents (methotrexate, sulfasalazine, or cyclosporin A) at the time of, and shortly before, the surgery (we set the standard as 1 month before the surgery). All RA and osteoarthritis (OA) patients met the clinical and radiographic criteria of the American College of Rheumatology. For immunohistochemistry and TSA-based immunofluorescent multiplex studies using human synovial biopsy tissue, the following patients were included: RA $n = 26$ (females, $n = 14$; high disease activity, $n = 19$, moderate disease activity, $n = 7$); OA $n = 21$ (females, $n = 12$). For the single-cell RNA sequencing, the following patients were included: RA $n = 5$ (females, $n = 5$, high disease activity $n = 5$). For the measurement of MDA, 8-OH-dG and iron levels in joint fluid, the following RA patients were included: $n = 20$ (females, $n = 16$; high disease activity, $n = 12$; moderate disease activity, $n = 8$). For the isolation of synovial fibroblasts from joint fluid and circulating fibrocytes from peripheral blood mononuclear cells, the following active RA patients were included: $n = 6$ (females, $n = 4$; high disease activity, $n = 3$; moderate disease activity, $n = 3$). Clinicopathological measurements were recorded, including disease duration, erythrocyte sedimentation rate (ESR), and general health or the patient's global assessment of disease activity. The Disease Activity Score 28 (DAS28) was calculated using the following formula: DAS28-ESR = 0.56 * sqrt (tender joint count) + 0.28 * sqrt (swollen joint count) + 0.70 * ln (ESR) + 0.014 * GH. GH, global health. The level of disease activity can be interpreted as low (DAS28-ESR ≤ 3.2), moderate (3.2 < DAS28-ESR ≤ 5.1), or as high disease activity (DAS28-ESR > 5.1). Detailed clinical characteristics were shown in Supplementary Tables 1–4.

**Single-cell RNA sequencing and data processing**. Synovial tissue samples were disaggregated into single-cell suspension as described. Briefly, specimens with connective tissues and fat removed were digested with collagenase II (4 mg/ml; Sigma, St Louis, MO, USA) in serum-free DMEM for at least one hour at 37 °C. The cell suspension was passed through nylon mesh, and the cells were then collected by centrifugation at $800 \times g$ for 5 min. Single-cell suspensions from disaggregated synovial tissues were assessed for cell quantity and cell viability using Trypan Blue. Single-cell RNA sequencing was performed using the Chromium (10X Genomics) instrument. Sample was run using the Chromium Single Cell 3′ Library & Gel Bead Kit v3 (10X Genomics). Gene expression libraries were sequenced using the NoveSeq 6000 (Illumina) by Gene Denovo Biotechnology Co., Ltd (Guangzhou, China). Raw gene expression matrices generated per sample using CellRanger (version 3.1.0) were combined in R (version 3.5.3) mapping to GRCh38, converted to a Seurat object using the Seurat R package (version 3.0.1). Seurat R package allows to easily explore QC metrics and filter cells[53]. All cells were removed that had either more than 20,000 UMIs, over 4000 or below 500 expressed genes, or over 10% UMIs derived from mitochondrial genome. T-SNE implemented in the "RunTSNE" functions with resolution sets to 0.5, respectively, were used to identify the deviations among all kinds of cells. Next, differential expression genes were determined using the Wilcoxon test implemented in the "FindAllMarkers" function, which was considered significant with an average natural logarithm (fold change) of at least 0.25 and a Bonferroni-adjusted $P$ value lower than 0.01.

**Tyramide signal amplification (TSA)-based immunofluorescent multiplex**. Multicolor immunofluorescence analyses were performed using 3-μm-thick

sections of formalin-fixed paraffin-embedded (FFPE) tissues. Briefly, slides were deparaffinized in xylene and hydrated in a series of decreasing graded ethanol series. After heat-induced antigen retrieval in citrate buffer (pH = 6), samples were permeabilized with 0.5% Triton X-100, blocked with 5% goat serum-phosphate-buffered saline (PBS), and sequentially co-stained with antibodies recognizing VCAM-1 (Abcam, ab134047), CD248 (Abcam, ab217535), 4-HNE (Abcam, ab46545), FAPα (Invitrogen, BMS168; Abcam, ab218164), Mfap4 (Thermo, PA5-24865), Sparcl1 (Santa Cruz, sc-514275), F4/80 (Cell Signaling Technology, #70076), rabbit IgG Isotype Control (Invitrogen, 31235), mouse IgG Isotype Control (Invitrogen, 14-4714-82), and DAPI. A TSA indirect kit (PerkinElmer) was used according to the manufacturer's instructions. Vectra® Polaris™ Imaging System (Akoya Biosciences) was used to collect images, and image analysis was performed using HALO™ Image Analysis Software.

**Flow cytometric analysis for ferroptosis sensitivity of main cell types in hyperplastic synovium**. Disaggregated single cells isolated from synovial biopsies were treated with GPX4 inhibitor RSL3 for 6 h before staining. Antibodies used were PerCP anti-CD3 (Biolegend, 317338), APC/Cyanine7 anti-CD31 (Biolegend, 303120), APC anti-CD68 (Biolegend, 333810), PE/Cyanine7 anti-CD19 (Biolegend, 302216), anti-FAPα (Invitrogen, BMS168), PE Goat anti-mouse IgG (Biolegend, 405307). Cell death was analyzed by SYTOX Green (Invitrogen) staining. The Fluorescence Minus One (FMO) controls were used to determine the cut-off point between background fluorescence and positive populations. The data was acquired with a BD Fortessa flow cytometer with BD FACSDiVa™ software and analyzed by FlowJo, v.10.5.3.

**Isolation and culture of cells from synovial tissues**. Rheumatoid arthritis synovial fibroblasts (RASFs) were isolated by enzymatic digestion from synovial tissues obtained during synovectomy. Single-cell suspensions from disaggregated synovial tissues were resuspended in DMEM supplemented with 10% fetal bovine serum (FBS) (Gibco, Grand Island, NY, USA). The harvested cells were cultured in 75 cm² culture flasks (Costar, Cambridge, MA, USA) with DMEM supplemented with 1% penicillin/streptomycin, 2% L-glutamine (Gibco), and 10% FBS at 37 °C in a humidified atmosphere of 5% $CO_2$. When the cells had grown to 80% confluence, they were detached with 0.25% trypsin, split in a 1:3 ratio, and reseeded in DMEM under the same conditions. To eliminate nonadherent cells, the plated cells were washed thoroughly with PBS. Isolated fibroblast-like synoviocytes (FLSs) were cultured in DMEM supplemented with 10% FBS. The cells used for our experiments were at passages 3–5 because these cells were more homogenous than were FLSs in the first and second passages and had better biological functions than the cells older than the fifth passage. The RASFs were identified by flow cytometric analysis as a homogeneous population (CD14⁻CD68⁻ Thy1⁺). The following antibodies were used to identify RASFs: PE anti-human CD14 (Biolegend, 367104), APC anti-human CD68 (Biolegend, 333810), and FITC anti-human CD90 (Thy1) (Biolegend, 328107). The data was acquired with a BD Fortessa flow cytometer with BD FACSDiVa™ software and analyzed by FlowJo, v.10.5.3.

**Isolation of circulating fibrocytes from human peripheral blood mononuclear cells**. PBMCs were isolated from human leukapheresis packs by Ficoll/Paque gradient centrifugation. Cells isolated in this way were then cultured for 2–3 days in DMEM supplemented with 20% FCS, penicillin, streptomycin, and L-glutamine. Typically, $2 \times 10^7$ PBMCs were plated into each T150 tissue culture flask. After 2–3 days, the media was aspirated from the PBMC culture, and the remaining adherent cells were washed with sterile PBS. Subsequently, the adherent cells were

harvested using Accutase (Innovative Cell Technologies Inc., San Diego, California, USA), which we found to be more efficient than the standard trypsin protocol for removing adherent cells from plastic surfaces. Previous data have indicated that the main contaminating cell types in these cultures are B lymphocytes (~3%), T lymphocytes (~13%), and monocytes/macrophages (up to 20% of the PBMC culture). Thus, we used immunomagnetic selection to deplete B lymphocytes (Dynabeads anti-CD19, Invitrogen, 11143D), T lymphocytes (Dynabeads anti-CD2, Invitrogen, 11159D), and monocytes/macrophages (Dynabeads anti-CD14, Invitrogen, 11149D) from our crude fibrocyte preparation. The remaining purified fibroblasts were returned to culture (DMEM supplemented with 20% FCS, penicillin, streptomycin, and L-glutamine) for an additional 5–7 days before analysis. Fibrocytes were routinely >90% pure, as determined by staining with PE anti-CD45 (Biolegend, 304008) and FITC anti-Collagen I (Sigma-Aldrich, FCMAB412F). The data was acquired with a BD Fortessa flow cytometer with BD FACSDiVa™ software and analyzed by FlowJo, v.10.5.3. Typically, our isolated fibrocytes represented ~0.5% of the overall pool of peripheral white blood cells.

**Quantification of GSH**. Cells were treated as indicated, and cellular GSH levels were assessed using the GSH-GSH/GSSG Ratio Detection Assay Kit (Abcam, ab138881) following the manufacturer's instructions. In brief, the GSH assay mixture was added to the whole-cell lysates for a one-step fluorometric reaction and incubated for 60 min while protected from light. Fluorescence was then monitored at EX/EM wavelengths of 490/520 nm, and GSH was calculated from the standard curve. The GSH concentration in each group was normalized to cell viability.

**Measurement of MDA in synovial fluid**. Synovial fluid samples were centrifuged at $1000 \times g$ for 10 min and filtered through a 0.2 μm filter to remove cells and debris. The intracellular MDA levels were assessed using the lipid peroxidation MDA Detection Kit (Beyotime, S0131M) following the manufacturer's instructions. In brief, a thiobarbituric acid (TBA) solution was incubated with synovial fluid samples and MDA standards at 100 °C for 15 min. The MDA-TBA adduct was quantified colorimetrically (OD = 532 nm).

**Measurement of 8-OHdG in synovial fluid**. 8-Hydroxy-2′-deoxyguanosine (8-OHdG) is a commonly used biomarker to assess oxidative stress. Levels of 8-OHdG in synovial fluid samples of RA patients were assessed by the 8-OHdG ELISA Kit (Abcam, ab201734) following the manufacturer's instructions.

**Measurement of iron level in synovial fluid**. Total iron levels in synovial fluid samples of RA patients were assessed by the Iron Assay Kit (Abcam, ab83366) according to the manufacturer's instructions. In brief, an iron reducer was added to the sample and standard wells. The mixture was incubated at 37 °C for 30 min, and iron was added and incubated for an additional 60 min. The output was then immediately assessed on a colorimetric microplate reader (OD = 593 nm).

**Western blotting**. Treated cells were harvested in radioimmunoprecipitation assay buffer (RIPA; Beyotime, P0013B) supplemented with protease inhibitor (Roche, 04693159001) and phenylmethylsulfonyl fluoride (PMSF; Beyotime, ST506) on ice, and the protein concentration was determined by using a BCA Protein Assay Kit (Thermo Scientific, 23250). Cellular proteins were separated via SDS-PAGE through 8–12% gels before they were transferred to polyvinylidene fluoride (PVDF) microporous membranes (EMD Millipore, IPVH00010). The blots were probed for 12 h at 4 °C with primary antibodies, and secondary antibodies were incubated with the PVDF membrane for 1 h at room temperature. Visualization was performed with a Molecular Imaging System (Carestream Health, INC., NY, USA). The antibodies used are as follows: anti-SLC40A1 rabbit antibody (Abcam, ab78066), anti-GCLM rabbit antibody (HuaBio, ET1705-87), anti-GCLC rabbit antibody (HuaBio, ET1704-38), anti-IκBα rabbit antibody (HuaBio, ET1603-6), anti-phospho-IκBα rabbit antibody (HuaBio, ET1609-78), anti-NF-kB p65 rabbit antibody (HuaBio, ET1603-12), anti-SLC7A11 rabbit antibody (Abcam, ab175186), anti-c-Jun rabbit polyclonal antibody (Proteintech, 24909-1-AP), anti-ferritin rabbit monoclonal antibody (Abcam, ab75973), anti-GAPDH rabbit antibody (HuaBio, R1210-1), and anti-β-actin antibody (HuaBio, 1102-1).

**RNA interference and transfection**. The small interfering RNA (siRNA) sequences were designed and synthesized by Shanghai GenePharma (Shanghai, China). The sequences were listed in Supplementary Table 5. The cells were transfected with the siRNAs using Lipofectamine 2000 (Invitrogen, Carlsbad, CA, USA).

**Real-time quantitative PCR assay**. For RT-PCR analysis, total RNA was isolated with an E.Z.N.A. Total RNA Kit II (OMEGA, BioTek, Norcross, USA) in RNase-free conditions from MSCs lysed in RNA lysis buffer and then reverse transcribed into cDNA with a Superscript First Strand Synthesis Kit (Invitrogen) according to the manufacturer's protocol. RT-PCR was then performed with a SYBR Green PCR kit (TaKaRa, Otsu, Japan) on the MxPro system to determine the expression levels of the genes of interest. All primers were synthesized by the Beijing Genomics

Institute (Shenzhen, China). The primer sequences are listed in Supplementary Table 6. RT-PCR was performed with the following conditions: 95 °C for 3 min; and 40 cycles of 95 °C for 15 s, 60 °C for 15 s, 72 °C for 15 s, and melting curve analysis. The $2^{-\Delta\Delta CT}$ method was used to quantify the relative expression of these genes.

**Measurement of cell death, cell viability, intracellular reactive oxygen species (ROS), and lipid peroxidation (lipid ROS)**. Cell death was analyzed by SYTOX Green (Invitrogen) staining followed by microscopy or flow cytometry. For cultured cells or 3D spheroids, cell viability was determined using the CellTiter-Glo™ Luminescent Cell Viability Assay or CellTiter-Glo 3D Cell Viability Assay (Promega) according to the manufacturer's instructions. Viability was calculated by normalizing ATP levels to those in control cells or spheroids. To analyze intracellular ROS, cells were stained with 10 μM ROS-sensitive probe 2′,7′-dichlorodihydrofluorescein diacetate (H2DCFDA, Invitrogen) for 1 h at 37 °C followed by flow cytometric analysis. To analyze lipid ROS, cells were stained with 5 μM BODIPY-C11 (Invitrogen) for 30 min at 37 °C followed by flow cytometric analysis.

**Generation of 3D spheroids**. Spheroids were generated by plating $10^3$ RASFs per well into U-bottom ultra-low adherence (ULA) 96-well plates (Corning). Optimal 3D structures were achieved by centrifugation at $600 \times g$ for 5 min followed by the addition of 2.5% (v/v) Matrigel (Corning). Plates were incubated for 72 h at 37 °C, 5% CO2, and 95% humidity to promote the formation of a single spheroid of cells. Spheroids were then treated with RSL3, IKE, and ferrostatin-1 in fresh medium containing Matrigel for the indicated times.

**Measurement of cellular labile iron pool (LIP)**. LIPs were measured via flow cytometry[54]. After trypsinization and two washes with 0.5 ml of PBS, the cells were incubated with 0.05 μM calcein-acetoxymethyl ester (AnaSpec) for 15 min at 37 °C. Then, the cells were washed twice with 0.5 ml of PBS and either incubated with deferiprone (100 μM) for 1 h at 37 °C or left untreated. The cells were analyzed with a flow cytometer. Calcein was excited at 488 nm, and fluorescence was measured at 525 nm. The difference in the mean cellular fluorescence with and without deferiprone incubation reflects the LIP levels.

**Mouse model of inflammatory arthritis**. Male DBA/1 mice aged 8–10 weeks were purchased from Beijing Vital River Laboratory Animal Technology Co., Ltd. (Beijing, China, stock # 218) and maintained in a specific pathogen-free (SPF) mice facility (room temperature, 20–22 °C; room humidity, 40–60%) with free access to food and water under a 12 h light/dark cycle. Experimental and control mice were co-housed in individual cages (maximum $n = 5$ per cage) during the experiment ensuring identical environmental conditions. All animal procedures were reviewed and approved by Laboratory Animal Ethics Committee of Fourth Military Medical University (IACUC-20191012). CIA was established by injecting DBA/1 mice with chicken type II collagen emulsified in complete Freund's adjuvant (CFA), followed by boosting 21 days later with type II collagen emulsified in incomplete Freund's adjuvant (IFA) (Chondrex). The development of arthritis was monitored and the arthritis score was evaluated every 2 days. The level of inflammation for each paw was graded from 0 to 4 by the following scale: 0 = absence of inflammation, 1 = paw with detectable swelling in a single digit, 2 = paw with swelling in more than one digit, 3 = paw with swelling of all digits and instep, and 4 = severe swelling of the paw and ankle. The arthritic scores of four paws were summed. For CIA model in Figs. 1 and 7, vehicle, IKE, liproxstatin-1, and/or etanercept were introduced into CIA mice via peritoneal injection at the moment of inflammation onset (defined as day 0). For CIA model in Supplementary Fig. 1g–j, liproxstatin-1 or vehicle was introduced into mice at day 7 after the first immunization (before inflammation onset). Vehicle for IKE and liproxstatin-1: 5% DMSO, 40% PEG300, 5% Tween80, 50% ddH2O. DBA/1 mice were monitored every 2 days for signs of arthritis based on paw swelling and arthritis scores. At the end of the study, blood samples were collected from mice under deep anesthesia, and then all the mice were euthanized by an intraperitoneal injection of pentobarbital.

**Mouse anti-chick type II collagen IgG antibody assay**. Anti-type II collagen antibody in serum of CIA mice was detected by Mouse Anti-Chick Type II Collagen IgG Antibody Assay ELISA Kit (Chondrex, 2031T) following the manufacturer's instruction.

**Micro-computed tomography (CT)**. Micro-CT was applied to mice to image the hind and front limbs. CT images were acquired using Inveon Acquisition Workplace from SIEMENS, Germany. Three-dimensional reconstruction was performed using the volume rendering method.

**Immunohistochemistry assay**. Immunohistochemical staining was performed using a streptavidin–peroxidase kit (ZSGB-Bio, China). The primary antibodies targeted the following proteins or modifications: 8-OHdG (Abcam, ab48508), 4-HNE (Abcam, ab48506, ab46545), FAPα (Abcam, ab53066), GPX4 (Abcam,

ab125066), GCLC (Abcam, ab53179), GCLM (Abcam, ab126704), p-NF-κB p65 (Thermo Fisher, 44-711G), rabbit IgG Isotype Control (Invitrogen, 31235), and mouse IgG Isotype Control (Invitrogen, 14-4714-82) with a standard avidin-biotin HRP detection system according to the instructions of the manufacturer (anti-mouse/rabbit HRP-DAB Cell & Tissue Staining Kit, R&D Systems, Minneapolis, MN). Tissues were counterstained with hematoxylin, dehydrated, and mounted. Quantitative analysis was performed with Image-Pro Plus Version 6.0 Software or HALO™ Image Analysis Software. For quantitative analysis using Image-Pro Plus, the percentage of positive cells was graded on a scale of 0 to 4 (0: negative, 1: 0–25%, 2: 26–50%, 3: 51–75%, 4: 76–100%). The staining intensity was further quantified from 0 to 3 (1: weak, 2: moderate, 3: intensive). The final score was obtained by multiplying the staining intensity score and percentage score.

**Cell invasion and migration assay**. For the cell invasion assay, $1.0 \times 10^5$ cells were suspended in serum-free DMEM. Cell suspensions (100 μl) were added to the upper compartment of the filter chamber, and the filter chamber was coated with 100 μl Matrigel (BD Bioscience, USA) at a 1:4 dilution ratio and inserted into a 24-well plate with DMEM, containing 20% fetal bovine serum. Culture dishes were incubated for 24 h at 37 °C in a 5% $CO_2$ atmosphere. The filters were then fixed with methanol for 10 min and stained with 0.5% methylrosaniline chloride for 30 min. A cotton swab was used to remove cells from the upper surface of the membrane. The cells on the back were counted at ×100 magnification in five random fields under a microscope. Each experiment was done in triplicate. The migration assay protocol was similar to the invasion experiment, except that no Matrigel was placed in the upper filter chamber.

**Statistics and reproducibility**. Unless otherwise stated, data are presented as mean ± SD. A two-tailed Student's t-test was used to compare means between two groups. For analysis of differences between the groups, one-way ANOVA for individual comparisons between the groups was performed. $IC_{50}$ values were calculated using nonlinear regression analysis (log(inhibitor) vs. response -- Variable slope (four parameters)). Statistical analysis was performed using GraphPad Prism 7.0 (GraphPad Prism Software, CA, USA). Differences were considered to be significant when $P < 0.05$. The western blot experiment was repeated independently for three times with similar results.

**Reporting summary**. Further information on research design is available in the Nature Research Reporting Summary linked to this article.

## Data availability

The scRNA-seq and RNA-seq data generated in this study have been deposited in the NCBI Sequence Read Archive (SRA) database under the accession code PRJNA725073 and PRJNA769634. The Croft et al. data used in this study are available in the GEO database under accession code GSE129087. All the source data and uncropped scans of blots supporting the findings of this study are provided with this paper as Source data file. A reporting summary is available as a Supplementary Information file. Source data are provided with this paper.

## Code availability

All codes have been deposited on Github (https://github.com/biolchen/TNF-and-Ferroptotic-Cell-Death).

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

## Acknowledgements
This work was supported by the National Natural Science Foundation of China 82022059 (to J.W.) and the Natural Science Basis Research Plan in Shaanxi Province of China 2019ZY-CXPT-03-01 (to P.Z.). This work was also supported by the Youth Science and Technology Nova Program of Shaanxi Province 2020KJXX-055 (to J.W.), and NCI Cancer Center Core Grant P30 CA008748 (to X.J.).

## Author contributions
J.W., P.Z., Z.-N.C., and X.J. conceived the original idea and designed the study. J.W., Z.F., L.C., Y.L., H.B., and J.-J.G. performed most experiments. Z.-H.Z., X.F., and Y.Z. collected human synovial biopsy tissue and performed tissue histology. Z.P., Y.Q., and L.Y. assisted with the CIA experimental arthritis model. K.W. performed micro-CT analysis. Q.H. and R.C. performed flow cytometry on human synovial tissue. G.N. assisted with the 3D model and performed microscope. P.Z., Z.-N.C., and X.J. supervised the research. J.W., Z.F., and L.C. wrote the paper.

## Competing interests
The authors declare no competing interests.
