## [Peer Review File · Nature Communications]

REVIEWER COMMENTS

Reviewer #1 (Remarks to the Author):

The paper by Wu et al, entitled 'Anti-TNF Therapy for Rheumatoid Arthritis Sensitizes Ferroptosis-Resistant Synovial Fibroblasts to Ferroptotic Cell Death' is very interesting a novel finding.

I do have some queries which authors should address.

- Can the authors provide clinical demographics or RA and OA patients, and of the high disease activity vs low disease activity RA.
- Could the authors clarify was this arthroplasty (ie end stage) or key-hole arthroscopy where biopsies were obtained. As above it would be important to include demographic and clarify medication data as the authors state that the RA patient had active disease, with mean duration of 10 years, but weren't on any steroid or second line medication which would be unusual with active disease and a disease duration of 10years?
- Figure 1A should quantify sub-lining vs lining layer as it would be interesting to see if expression is more localised to fibroblasts in a specific anatomical location.
- Should show RA images of High vs Low disease activity for MDA with IgG controls
- Similarly did the authors stain high vs low disease activity for 4HNe and 8-oxdG
- How did the author define High disease activity vs low disease activity
- Figure 1F -should quantify Fab and F4/F80
- Line 130 the authors state ' Interestingly, the surviving FAP α + fibroblast cells that evaded IKE-induced ferroptosis were mainly surrounded by macrophages (Extended Data Fig. 1h), implying that macrophages protect fibroblasts against lipid oxidative stress and ferroptotic death' . Do the authors know if these macrophages are the recently described protective CX3CR1 macrophages in the RA joint.
- For Histology Authors need to include IgG controls throughout.
- Were the Fib-a clusters or Fib-b clusters Thy1 + or Thy1-.
- Were FMO controls used for flow cytometry
- The authors used cellular communication networks to examine potential receptor-ligand pair interactions between macrophages and Fibroblasts. There is a spectrum of macrophages in the joint, did the author identify specific macrophages subtypes associated with these interactions
- For the Analysis of the potential receptor-ligand pair interactions can the authors clarify if the analysis was intra-analysis ie they examined the interaction of macrophages with Fib-a and Fib-b within each RA patient, and then combined analysis.
- In figure 4 the authors show that TNF- α protects RA fibroblasts from ferroptosis by enhancing the GSH biosynthetic pathway via activation of NF- κ B. The manuscript would be greatly enhanced if the authors could show some functional outputs of the fibroblasts in response to the various treatments (RSL3; si GCLC, SIN3K) including fibroblasts invasiveness, migration capacity and secretion of key MMPs and pro-inflammatory mediators
- In the last figure the author show that TNF- α antagonist sensitizes RA fibroblasts to ferroptosis induction in CIA model mice. Similar histological staining in Human RA tissue pre/post TNFi would demonstrate the translational aspect of the study.

Reviewer #2 (Remarks to the Author):

The manuscript by Jiao Wu and colleagues examines alleviation of RA through activation of ferroptosis in synovial fibroblasts and the role of TNF α as a driver of ferroptosis resistance in a subset of the cells. This is an interesting story which, I think, convincingly suggest that synovial fibroblasts may have elevated susceptibility to ferroptosis inducers in the RA patients. It also demonstrates that a subset of fibroblasts may be protected from ferroptosis induced by an exogenous trigger through TNF-

dependent signaling. At the same time, I don't think that the manuscript in the current form is sufficiently impactful for the publication in Nature Communications. First, I don't see evidence that ferroptosis is playing a significant role in RA in the absence of IKE treatment, for example there is no difference in scores in Fig. 1d-e in CIA and CIA+Lip-1 group. Therefore, impact of the findings is dependent on the feasibility of using ferroptosis inducers in the context of RA. Second, while activation of ferroptosis may be a good strategy in more acute settings, I find it questionable that such approach will be productive in a chronic and non-life threatening disease like RA, including long term death-associated inflammation, local tissue damage and damage to a variety of other tissues. To make discovery of the effect of ant-TNFs more impactful, the authors need to convince the readers first that exogenous activation of ferroptosis is a viable approach to RA. Therefore, I would recommend the manuscript in its present form for consideration in a more specialized journal.

Reviewer #3 (Remarks to the Author):

The concepts introduced in the study are novel and interesting, and there is a wealth of data supporting their hypothesis. However, there are some issues that should be addressed. Overall the manuscript is extremely dense and each figure has up to 20 sections. The authors should focus their studies on the key data required to make their case and prove the mechanism and eliminate speculation and conclusions based on assumptions like much of Figure 2 and negative data related to the other cytokines. A shorter manuscript will be much more accessible and understandable.

1. A previous study reported that TNF stimulation induces NF- κ B activation through TNFR1/PKC α /IKK α/β , results in p38 MAPK-, JNK-1/2-dependent NOX/ROS pathways in RA SFs (Mediators Inflamm. 2014;2014:279171). Since NOX/ROS induce lipid oxidation and ferroptosis, this could conflict with the present study. The authors should discuss difference between the studies.
2. Line 130: The authors should quantify the contention that surviving FAP+ fibroblasts are more likely to be surrounded by macrophages.
3. The data from Figure 2l through 2r are primarily computational and from biologic studies. Comments like "TNF signaling through NF κ B was enriched in fibroblast clusters 1 and 2", "Macrophages also initiated TGF β signaling on fibroblasts", "TNF released from macrophages showed a higher interaction with TNFR2 in Fib a than in Fib b" and many others are inferences rather than the result of biologic validation. The text could be significantly shortened and the comments here and throughout the manuscript should distinguish between computational predictions, potential associations based on informatics and biological validation. The authors should test some of these predictions and present data supporting them.
4. The fact that TNF, IL-1 and IL-6 had different effects on ferroptosis provides a way to dissect the signaling mechanisms. The authors conclusion that NF κ B is the primary mechanism related to the TNF effect seems unlikely because IL-1 also induces the same transcription factor. A more detailed signaling study defining how TNF leads to GSH production would be important. If it is through NF κ B, than methods to prevent NF κ B activation would be an important step to proving their hypothesis.
5. The translational approach of testing etanercept and IKE is interesting and important, although they have not shown whether the effects are due to modulating adaptive or innate immune mechanisms. For example, what was the effect on type II collagen antibodies? The authors should test their treatment regimen in established disease (typically day 35 or later in mouse CIA) or in a model that is not dependent on adaptive immunity such as a serum transfer model to determine the mechanism of action. Prolonged treatment of mice with human etanercept leads to neutralizing antibodies and occurs within a couple weeks. This issue can be avoided with shorter term studies.
6. RSL3 concentrations used in these experiments were 0.125 μ M. Others, however, require higher concentration (2 μ M, 3 μ M, and 5 μ M) than those (Oncogene. 2017 Oct 5;36(40) 5593-5608; Cell Death Diff. 2020 Dec 17; Front Pharmacol. 2018 Nov 22;9:1371). Prior to do in vitro cell experiment with IKE, RSL3, and others, the author should perform dose responses to determine the optimal concentrations.
7. The manuscript should be reviewed by a statistician to assess the methods, especially Fig. 1d, e,

and k and Fig. 3a.

8. Minor points: in line 110, the authors comment on FAP+ fibroblasts in RA synovium and cultured RA synovial fibroblasts. As noted by several authors, cultured SF are almost all FAP+/CD90+/podoplanin+ and have a phenotype that is a combination of lining and sublining fibroblasts. Also, the title should be modified because the authors do not actually test their hypothesis in patients with RA.

The figures needed to address each concern were combined and presented as Response Figures.

Reviewer	Question	Response
REVIEWER 1	1. Clinical demographics of RA and OA patients, disease activity of RA.	Detailed clinical demographics updated. Supplementary Table 1-4
	2. Why RA patient with active disease weren't on any steroid or second line medication.	Detail of patients clarified.
	3. Quantification of 4-HNE in sub-lining vs lining layer.	Response Figure 1
	4. Statistics of MDA, 4-HNE and 8-OHdG of RA in high vs low disease activity.	Response Figure 2
	5. Definition of high, moderate and low disease activity in RA.	Formula for DAS28-ESR and standards for disease activity added.
	6. Quantification of FAP and F4/80 in fibroblasts.	Response Figure 3
	7. Explore CX3CR1+ macrophage in the RA joint.	Response Figure 4
	8. Provide all IgG controls of IHC assay.	Response Figure 5
	9. Analyze the expression of THY1 in fibroblast clusters in RNA-seq.	Response Figure 6
	10. FMO control in Flow Cytometry assay.	Response Figure 7
	11. Analyze specific macrophage subtypes interacts with fibroblast in RNA-seq.	Response Figure 8
	12. Clarify if the interaction analysis was within each RA patient and combined.	Response Figure 9
	13. Detect the invasiveness, migration capacity, secretion of key MMPs and pro-inflammatory mediators of the fibroblasts in response to various treatments.	Response Figure 10
	14. Human RA tissue pre/post TNFi	Difficult was explained.
REVIEWER 2	1. Demonstrate if ferroptosis plays an important role in RA without IKE treatment, such as CIA and CIA+Lip-1 group.	Response Figure 11
	2. Question that whether ferroptosis will be productive in a chronic and non-life-threatening disease like RA, including long term death-associated inflammation and local tissue damage and damage to a variety of other tissues.	Response Figure 12
REVIEWER 3	1. Discuss the difference between this paper and a previous study about TNF-NOX signaling.	Response Figure 13
	2. Quantification the contention between survival FAP+ fibroblasts and macrophages.	Response Figure 14
	3. Some data are redundant. Validate some computational inferences like "TNF released from macrophages showed a higher interaction with TNFR2 in Fib a than in Fib b".	Redundant data were deleted. Response Figure 15
	4. Prevent NFkB activation to prove hypothesis.	Response Figure 16

	5. The modulation of adaptive or innate immune in CIA model treated with IKE and etanercept.	Response Figure 17
	6. In vitro cell experiment perform dose responses with IKE and RSL-3.	Response Figure 18, Response Figure 19
	7. The manuscript should be reviewed by a statistician to assess the methods.	All data were reviewed. Response Figure 20
	8. Modify the title and comment about the characteristics of RA synovial fibroblasts.	Title and comment were modified.

REVIEWER 1

The paper by Wu et al, entitled 'Anti-TNF Therapy for Rheumatoid Arthritis Sensitizes Ferroptosis-Resistant Synovial Fibroblasts to Ferroptotic Cell Death' is very interesting a novel finding.

Response: We thank the reviewer for the positive and encouraging assessment. I do have some queries which authors should address.

- Can the authors provide clinical demographics or RA and OA patients, and of the high disease activity vs low disease activity RA.

Response: We thank the reviewer for this suggestion. We sincerely apologized for our negligence about the clinical demographics. The detailed and updated clinical demographics for RA and OA patients for different experiments were collected before and are provided in the revision (Supplementary Table 1-4). The DAS28-ESR of every RA patient was shown in source data for tables. In brief, for immunohistochemistry and TSA-based immunofluorescent multiplex studies using human synovial biopsy tissue, the following patients were included: RA n = 26 (females, n = 14; high disease activity, n = 19, moderate disease activity, n = 7); OA n = 21 (females, n = 12). For the single-cell RNA sequencing, the following patients were included: RA n = 5 (females, n = 5, high disease activity n = 5). For the measurement of MDA, 8-OH-dG and iron levels in joint fluid, the following RA patients were included: n = 20 (females, n = 16; high disease activity, n = 12; moderate disease activity, n = 8). For the isolation of synovial fibroblasts from joint fluid and circulating fibrocytes from peripheral blood mononuclear cells, the following active RA patients were included: n = 6 (females, n = 4; high disease activity, n = 3; moderate disease activity, n = 3).

- Could the authors clarify was this arthroplasty (ie end stage) or key-hole arthroscopy where biopsies were obtained. As above it would be important to include demographic and clarify medication data as the authors state that the RA patient had active disease, with mean duration of 10 years, but weren't on any steroid or second line medication which would be unusual with active disease and a disease duration of 10years?

Response: We thank the reviewer for this comment. The biopsies were obtained from

key-hole arthroscopy (during arthroscopic cleansing of the knee) and the synovial fluid samples were obtained during joint aspiration. Both arthroscopic cleansing of the knee and joint aspiration are used as routine treatments for therapeutic reasons, which help to remove damaged or infected tissue and to reduce the inflammation. In our original manuscript, we stated that “No patients included in the study *were being* treated with corticosteroids or second-line drug agents”, which we intend to mean they were not under these treatments at the time of, and shortly before, the surgery (we set the standard as one month before the surgery). In clinic, a lot of rheumatoid arthritis patients stopped taking their medications within two years after they started them when it went into remission. However, there is no cure for rheumatoid arthritis. Thus, there are a lot of patients with relapsed active disease who don't receive treatments for a not long term before the therapeutic surgery. We apologize for our lack of clarity, leading to the misunderstanding that the patients were not under treatment for the whole period since their diagnosis, which is not the case. We have made this clear in the revision.

- Figure 1A should quantify sub-lining vs lining layer as it would be interesting to see if expression is more localised to fibroblasts in a specific anatomical location.

Response: We agree this is an important question to address. In Figure 1a of original version (revision Fig. 1d), we used MDA Detection Kit to measure MDA concentration in joint fluid, not IHC. I guess the reviewer means the staining of 4-HNE (revision Fig 1b). In the revised version, we did TSA-based immunofluorescent multiplex assay and quantify the sub-lining vs lining layer as the reviewer suggested. Fibroblasts of lining layer was stained with VCAM-1 and fibroblasts of sub-lining layer was stained with CD248⁺. We found the staining of 4-HNE remained unchanged in the fibroblasts of lining layer and sub-lining layer (revision Fig. 1c).

Response Figure 1

- Should show RA images of High vs Low disease activity for MDA with IgG controls

Response: We thank the reviewer for this suggestion. In the original version Figure 1a (revision Fig. 1d), we already measured MDA concentration in joint effusion of RA patients with high or moderate disease activity with a MDA Detection Kit, which is not a IHC result. To avoid misunderstanding and distinguish the results from synovium and joint effusion, we labeled each plot in the revision.

- Similarly did the authors stain high vs low disease activity for 4HNE and 8-oxdG

Response: We thank the reviewer for this suggestion. Since RA patients with low disease activity usually do not meet the indications for surgery or joint aspiration, there're not enough samples from patients with low disease activity. All patients involved in our study are of high or moderate disease activity. In the original version, 8-OHdG was measured by ELISA and the concentration in the joint fluid of RA patients with high and moderate disease activities was shown in Extended Data Figure 1b (revision Supplementary Fig. 1e). In the revision, 8-OHdG was detected by IHC in RA and OA synovium. The quantification in RA and OA, and in high vs moderate disease activity of RA was shown. In addition, we added the quantification in high vs moderate disease activity for 4HNE staining. Hyperplastic rheumatoid synovium appears to show higher levels of 8-OHdG (revision Fig. 1a) and 4-HNE (revision Fig. 1b) compared with that of OA patients. However, no significant difference in the staining of 8-OHdG and 4-HNE was found between the patients with high and moderate disease activity (revision Supplementary Fig. 1b-c).

Response Figure 2

- How did the author define High disease activity vs low disease activity

Response: The standard for defining high and moderate disease activities was added to the METHODS section. In brief, clinicopathological measurements were recorded,

including disease duration, erythrocyte sedimentation rate (ESR) and general health or the patient's global assessment of disease activity. The Disease Activity Score 28 (DAS28) was calculated using the following formula: $\text{DAS28-ESR} = 0.56 * \text{sqrt}(\text{tender joint count}) + 0.28 * \text{sqrt}(\text{swollen joint count}) + 0.70 * \ln(\text{ESR}) + 0.014 * \text{GH}$. GH, global health. The level of disease activity can be interpreted as low ($\text{DAS28-ESR} \leq 3.2$), moderate ($3.2 < \text{DAS28-ESR} \leq 5.1$), or as high disease activity ($\text{DAS28-ESR} > 5.1$).

- Figure 1F -should quantify Fab and F4/F80

Response: We think the reviewer means Figure 1i (revision Fig. 2b). We quantified FAP and F4/80 as suggested using HALO™ Image Analysis Software. IKE treatment strikingly decreased the population of FAP α + fibroblasts (revision Fig. 2b). Although IKE seemed to increase the mean fluorescence intensity of F4/80+ macrophage, the Cohen's d is only 0.21, which indicates a small effect size (revision Fig. 2b).

Response Figure 3

revision Fig. 2b

- Line 130 the authors state 'Interestingly, the surviving FAP α + fibroblast cells that evaded IKE-induced ferroptosis were mainly surrounded by macrophages (Extended Data Fig. 1h), implying that macrophages protect fibroblasts against lipid oxidative stress and ferroptotic death'. Do the authors know if these macrophages are the recently described protective CX3CR1 macrophages in the RA joint.

Response: We thank the reviewer for this insightful suggestion and have performed additional experiments accordingly. The epithelial-like CX3CR1⁺ macrophages restrict the inflammatory reaction of RA and protect joint structures by providing a tight-junction-mediated shield at the synovial lining for intra-articular structures (*Nature*. 2019 Aug;572(7771):670-675). In our study, the macrophages surrounding the survival fibroblasts that evaded IKE-induced ferroptosis give rise to the assumption that macrophages may protect fibroblasts against lipid oxidative stress and ferroptotic cell death. Although the mechanisms seem to be different, we agree it would be interesting to figure out if these macrophages also express CX3CR1. We co-stained the CX3CR1-expressing macrophages and FAP α ⁺ fibroblasts to figure out if the macrophages surrounded are CX3CR1 positive (anti-CX3CR1, Abcam, ab8021). The number of CX3CR1⁺F4/80⁺ macrophages within the 50 μ m radius of surviving FAP α ⁺ fibroblasts that evaded IKE-induced ferroptosis significantly increased in joints of IKE-treated CIA mice (**Response Figure 4**). However, the percentage of CX3CR1⁺ F4/80⁺ cells among total F4/80⁺ macrophages remained unchanged. Although these data didn't indicate specific enrichment of CX3CR1⁺ macrophages around survival fibroblasts, it's worthy to explore if there are specific macrophages clusters playing a key role in the protective effect against ferroptosis.

Response Figure 4

- For Histology Authors need to include IgG controls throughout.

Response: We thank the reviewer for this suggestion. For all IHC staining and Tyramide signal amplification (TSA)-based immunofluorescent multiplex, we did IgG controls. Since all the antibodies used for normal IHC and TSA-based immunofluorescent multiplex are produced by mouse or rabbit, representative images were shown in the revision. We showed immunohistochemistry on formalin-fixed paraffin embedded human RA synovium or mouse CIA joints using 10 µg/mL of rabbit IgG1 isotype control or mouse IgG1 isotype control followed by an anti-mouse/rabbit avidin-biotin HRP detection system in **revision Supplementary Fig. 1a**. The IgG controls for PTGS2 and GPX4 in mouse CIA joints were shown in **revision Supplementary Fig. 1k**. Fluorescent multiplex IHC staining of mouse CIA joints using 10 µg/mL of rabbit IgG1 isotype control or mouse IgG1 isotype control followed by TSA detection kit was shown in **revision Supplementary Fig. 1d**.

Response Figure 5

- Were the Fib-a clusters or Fib-b clusters Thy1+ or Thy1-.

Response: We checked our single cell RNA-seq data. We found the expression of Thy1 showed no difference between Fib-a cluster and Fib-b cluster (**Response Figure**

6).

Response Figure 6

- Were FMO controls used for flow cytometry

Response: For cell death staining with SYTOX and lipid ROS staining with BODIPY, the flow strategy was shown in supplementary file. For detection of cell death (SYTOX staining) of main cell types in hyperplastic synovium treated with ferroptosis inducer, we used single staining controls for multicolor flow cytometry experiments to determine the levels of compensation. To better determine the cut-off point between background fluorescence and positive populations, we redid this experiment and the data was updated. The Fluorescence Minus One (FMO) controls were added (revision Supplementary Fig. 2a-b).

Response Figure 7

- The authors used cellular communication networks to examine potential receptor-ligand pair interactions between macrophages and Fibroblasts. There is a spectrum of macrophages in the joint, did the author identify specific macrophages subtypes associated with these interactions

Response: We agree this is an important question to address. We tried to identify specific macrophages subtypes and analysis the receptor-ligand interactions between different macrophage clusters with fibroblast clusters. Primary cell cluster analysis was performed using Seurat's FindClusters function (resolution=0.5). This analysis led to the identification of 7 main macrophages clusters. The macrophages of the 7 clusters were combined, and the results of 18 clusters were output with the threshold of resolution =0.5, and the results of 3 clusters were output with the threshold of resolution=0.08. Based on the above three methods for clustering, cell-cell interactions between macrophages clusters and fibroblasts clusters were identified using CellPhoneDB. Subsequently, the ligand-receptor relationships among them were identified. **Response Figure 8a-c** revealed the numbers of significant ligand-receptor pairs among the predicted cell types. **Response Figure 8d-f** listed the means of the average level of TNF-TNFRSF1B interaction. Although the mean of TNF-TNFRSF1B interaction is higher between each macrophage cluster with Fib a than with Fib b, no specific cluster of macrophages showed more significant interaction with Fib a than other clusters.

Response Figure 8

Response Fig. 8d

Subsets	TNF_TNFRSF1B	
	mean	p value
macrophages-0 Fib a	1.406	0
macrophages-0 Fib b	1.321	0
macrophages-1 Fib a	0.738	1
macrophages-1 Fib b	0.652	1
macrophages-2 Fib a	0.429	1
macrophages-2 Fib b	0.343	1

Response Fig. 8e

Subsets	TNF_TNFRSF1B	
	mean	p value
macrophages-0 Fib a	1.076	0.65
macrophages-0 Fib b	0.889	0.992
macrophages-1 Fib a	2.938	0
macrophages-1 Fib b	2.852	0
macrophages-10 Fib a	0.836	0.996
macrophages-10 Fib b	0.75	1
macrophages-11 Fib a	0.267	1
macrophages-11 Fib b	0.481	1
macrophages-12 Fib a	0.647	1
macrophages-12 Fib b	0.56	1
macrophages-13 Fib a	1.59	0.008
macrophages-13 Fib b	1.503	0.015
macrophages-14 Fib a	0.394	1
macrophages-14 Fib b	0.308	1
macrophages-15 Fib a	0.562	1
macrophages-15 Fib b	0.476	1
macrophages-16 Fib a	0.736	0.908
macrophages-16 Fib b	0.649	0.988
macrophages-17 Fib a	0	1
macrophages-17 Fib b	0	1
macrophages-2 Fib a	0.605	1
macrophages-2 Fib b	0.518	1
macrophages-3 Fib a	0.804	1
macrophages-3 Fib b	0.717	1
macrophages-4 Fib a	0.544	1
macrophages-4 Fib b	0.458	1
macrophages-5 Fib a	0.382	1
macrophages-5 Fib b	0.296	1
macrophages-6 Fib a	0.527	1
macrophages-6 Fib b	0.441	1
macrophages-7 Fib a	0.514	1
macrophages-7 Fib b	0.427	1
macrophages-8 Fib a	0.441	1
macrophages-8 Fib b	0.355	1
macrophages-9 Fib a	0.67	1
macrophages-9 Fib b	0.583	1

Response Fig. 8c

Response Fig. 8f

Subsets	TNF_TNFRSF1B	
	mean	p value
macrophages-11 Fib a	0.432	1
macrophages-11 Fib b	0.346	1
macrophages-15 Fib a	0.541	1
macrophages-15 Fib b	0.454	1
macrophages-18 Fib a	0.388	1
macrophages-18 Fib b	0.301	1
macrophages-2 Fib a	0.536	1
macrophages-2 Fib b	0.449	1
macrophages-3 Fib a	0.752	1
macrophages-3 Fib b	0.695	1
macrophages-4 Fib a	0.891	1
macrophages-4 Fib b	0.804	1
macrophages-0 Fib a	3.394	0
macrophages-5 Fib b	3.306	0

- For the Analysis of the potential receptor-ligand pair interactions can the authors clarify if the analysis was intra-analysis ie they examined the interaction of macrophages with Fib-a and Fib-a within each RA patient, and then combined analysis.

Response: In the analysis of cell–cell communications and receptor-ligand pair interactions, all RA patients were considered as a group. Here we listed some examples from published articles that performed the same strategy, indicating that it’s widely acceptable to consider the individuals as a group during cell–cell communications analysis (**Response Figure 9**).

(1) *Nature Biotechnol.* 2020;38(8):970-979_Fig. 4: Heat map depicting cell–cell communications between all identified cell types derived from log-scaled ligand–receptor interaction counts in **moderate group** (left, n = 8 patients and 14 samples) and **critical group** (right, n = 11 patients and 13 samples) patients.

(2) *Nature communications* 2021;12:3709_Fig. 4 : Heatmap showing the numbers of inter-populations communications with each other in **normal scars group** (a) and in **keloid tissues group** (b).

(3) *Circulation.* 2020;142(15):1448-1463_Fig. 6: Chord plot summarizing interconnections between different cardiac cell types from hearts of **control mice**.

Response Figure 9

Nature Biotechnol. 2020;38(8):970-979_Fig. 4

Nature communications 2021;12:3709_Fig. 4

Circulation. 2020;142(15):1448-1463_Fig. 6

- In figure 4 the authors show that TNF- α protects RA fibroblasts from ferroptosis by enhancing the GSH biosynthetic pathway via activation of NF- κ B. The manuscript would be greatly enhanced if the authors could show some functional outputs of the fibroblasts in response to the various treatments (RSL3; si GCLC, SINFKB) including fibroblasts invasiveness, migration capacity and secretion of key MMPS and pro-inflammatory mediators

Response: We agree that functional outputs of fibroblasts would greatly improve the significance of ferroptosis induction therapy. Since lethal concentration of ferroptosis inducer would directly result in ferroptotic cell death, which makes invasion and migration assay impossible, fibroblasts were transfected with scramble siRNA or siRNA targeting NF- κ B p65, and then treated with sublethal concentration of RSL3. We found that sublethal concentration of RSL3 resulted in markedly reduced invasion and migration capacities and decreased MMP8, MMP9, MMP11, MMP13 and MMP14 expression in TNF- α -induced RA fibroblasts with NF- κ B p65 subunit knockdown (revision Supplementary Fig. 10e-h). These results indicated a critical contribution of sublethal doses of ferroptosis-inducers to the modulation of the aggressive behaviors of RA fibroblasts.

Response Figure 10

revision Supplementary Fig. 10e

revision Supplementary Fig. 10f

revision Supplementary Fig. 10g

revision Supplementary Fig. 10h

- In the last figure the author show that TNF- α antagonist sensitizes RA fibroblasts to ferroptosis induction in CIA model mice. Similar histological staining in Human RA tissue pre/post TNFi would demonstrate the translational aspect of the study.

Response: We agree such data from clinical samples will be very helpful. However, practically it is not feasible for us to obtain RA tissues from patients receiving arthroscopic cleansing of the knee, both before and after anti-TNF treatment without therapeutic reasons.

REVIEWER 2

The manuscript by Jiao Wu and colleagues examines alleviation of RA through activation of ferroptosis in synovial fibroblasts and the role of TNFa as a driver of ferroptosis resistance in a subset of the cells. This is an interesting story which, I think, convincingly suggest that synovial fibroblasts may have elevated susceptibility to ferroptosis inducers in the RA patients. It also demonstrates that a subset of fibroblasts may be protected from ferroptosis induced by an exogenous trigger through TNF-dependent signaling. At the same time, I don't think that the manuscript in the current form is sufficiently impactful for the publication in Nature Communications. First, I don't see evidence that ferroptosis is playing a significant role in RA in the absence of IKE treatment, for example there is no difference in scores in Fig. 1d-e in CIA and CIA+Lip-1 group. Therefore, impact of the findings is dependent on the feasibility of using ferroptosis inducers in the context of RA.

Response: We truly appreciate the reviewer' valuable comments and suggestions. First, as we demonstrated in Figure 1, hyperplastic rheumatoid synovium and synovial fluid of RA patients showed increased levels of iron and lipid peroxidation, indicate an increased sensitivity to ferroptosis induction but not necessarily ongoing ferroptosis in fibroblasts. In this sense, the increased ferroptosis tendency without actual ongoing ferroptosis is exactly the basis of our proposed ferroptosis induction-based therapy. The reviewer is correct to conclude that such therapy depends on the availability/feasibility of clinical ferroptosis inducers. However, to conduct mechanistic and preclinical animal model investigation – the goal of this paper – is

crucial to prove this concept and to define the underlying mechanism, which will set the stage for the development of ferroptosis-inducing clinical agents. We also agree that the role of lipid peroxidation and increased ferroptosis tendency in the entire progression of RA development needs to be explored. In our previous experiment, liproxstatin-1 was administrated to CIA mice that already developed active inflammation and failed to suppress the development of inflammation and joint damage (revision Fig. 1f and 1g). In the revision, we started liproxstatin-1 treatment from an earlier time point, which helps us to better investigate the role of lipid peroxidation in different stages of RA development (revision Supplementary Fig. 1g). We found liproxstatin-1 treatment started at the pre-symptomatic stages of RA (at day 7 after the first immunization) constantly prevented the shown up of significant arthritis symptoms and the development of joint inflammation (revision Supplementary Fig. 1h-i) and joint destruction (Supplementary Fig. 1j). High levels of lipid peroxidation and 4-HNE expression correlate with low in vivo oxygen tension levels, and animal studies show that hypoxia occurs at the pre-arthritic stage of RA (*Clin Exp Rheumatol.* 2008;26(4):646-8). In addition, the modification of proteins by the cytotoxic 4-HNE may contribute to the onset of autoimmune reactions or even autoimmune disease process (*Autoimmun Rev.* 2008;7(7):567-73). Taken together, it's reasonable to hypothesis the considerable contribution of lipid peroxidation to the onset of RA pathogenesis.

Response Figure 11

Second, while activation of ferroptosis may be a good strategy in more acute settings, I find it questionable that such approach will be productive in a chronic and non-life threatening disease like RA, including long term death-associated inflammation, local tissue damage and damage to a variety of other tissues. To make discovery of the effect off anti-TNFs more impactful, the authors need to convince the readers first that exogenous activation of ferroptosis is a viable approach to RA. Therefore, I would recommend the manuscript in its present form for consideration in a more specialized journal.

Response: We thank the reviewer for bringing up this crucial concern. We truly understand the reviewer's concern about the potential adverse side effects of the ferroptosis-inducing therapy. Indeed, this is always a major consideration for any potential therapy and can only be formally tested by a clinical trial, which is outside the scope of this paper. We agree that RA is a non-life threatening disease. However, current therapies are unable to completely prevent progressive joint erosion and cure this disease. Furthermore, corticosteroids or second-line drug agents share a similar therapeutic response ceiling. Thus, we believe that novel therapies need to be developed. In addition to RA, there are other diseases related to abnormal activation of fibroblasts, such as lung fibrosis (which is life-threatening), systemic sclerosis and liver fibrosis. In tumor microenvironment, the activated cancer-associated fibroblasts contribute to the regulation of tumor occurrence, development, metastasis, and therapeutic resistance, and emerged has a prominent target in anti-cancer therapy. Considering the contribution of fibroblasts to a variety of diseases, targeted strategies focusing on interfering with the function or survival of fibroblasts deserve further study. In our previous article, we found that fibroblasts and mesenchymal cells are sensitive to ferroptosis induction, which inspires us about the possibility of targeting activated fibroblasts with ferroptosis inducer as a kind of therapeutic strategy.

Specific to our case, we tend to think SLC7A11 (the target of IKE) is a promising drug target that might be well tolerated by normal tissues, for the following reasons. First, mice with genetic knockout of SLC7A11 gene develop normally and show very minor phenotype for normal organs as adults (Hepatology. 2017;66(2):449-465), indicating low on-target adverse side effects. Second and specific to IKE as a proto-drug, it has been widely tested as an *in vivo* ferroptosis-inducing cancer therapeutic agent (*Cell Chemical Biology*, 2019, 26(5):623-633.e9. *Nature*, 2019;572(7769):402-406. *ACS Chem Biol*. 2020; 15(2): 469–484) because of its

strong potency and metabolic stability *in vivo*. The safety of IKE was also demonstrated in these studies (although in short term with high doses which is more relevant to cancer treatment).

Further, we collected the main organs, including heart, liver, spleen, lung and kidney of ferroptosis inducer (IKE)-treated mice (IKE 20mg/kg, twice a week, endpoint >57 days from CIA induction). We did H&E staining and didn't find significant pathological changes (revision Supplementary Fig. 12a). Although we didn't observe obvious pathological changes in main organs of CIA mice treated with low dose IKE, it's noteworthy that long-term administration of ferroptosis inducers may increase the risk of death-associated inflammation and damage to normal tissues. Activated fibroblasts showed higher sensitivity to ferroptosis inducers compared to other main cell types in the hyperplastic synovium, which partially increases the specificity of ferroptosis induction. In addition, fibroblasts-directed ferroptosis strategies should be developed. Future studies will potentially identify surface proteins that are more specific to fibroblasts, which help to facilitate the ferroptosis strategies targeting fibroblasts therapeutically. Moreover, two FDA-approved rheumatoid arthritis drugs, sulfasalazine and auranofin, were also proven to effectively trigger ferroptosis (*Signal Transduct Target Ther.* 2020 Jul 31;5(1):138. *Cancer Res* 2021;81:1896–908. *Oncol Rep.* 2019 Aug;42(2):826-838). The combination of these drugs together with anti-TNF α therapy could be tested to reduce the risk of ferroptosis induction of proto-drug.

Response Figure 12

revision Supplementary Fig. 12a

REVIEWER 3

The concepts introduced in the study are novel and interesting, and there is a wealth of data supporting their hypothesis. However, there are some issues that should be addressed. Overall the manuscript is extremely dense and each figure has up to 20 sections. The authors should focus their studies on the key data required to make their case and prove the mechanism and eliminate speculation and conclusions based on assumptions like much of Figure 2 and negative data related to the other cytokines. A shorter manuscript will be much more accessible and understandable.

Response: We thank the reviewer for the positive and encouraging assessment. For the presentation issue the reviewer pointed out, we revised the manuscript to make it less dense and flow better. The negative data related to IL1 β and some assumptions based on single cell RNA sequencing were deleted. In addition, more experiments were added to prove the key mechanisms.

1. A previous study reported that TNF stimulation induces NF- κ B activation through TNFR1/PKC α /IKK α / β , results in p38 MAPK-, JNK-1/2-dependent NOX/ROS pathways in RA SFs (*Mediators Inflamm.* 2014;2014:279171). Since NOX/ROS induce lipid oxidation and ferroptosis, this could conflict with the present study. The authors should discuss difference between the studies.

Response: We agree this is an important question to address. We found that TNF- α enhances the cystine uptake and cellular GSH biosynthetic pathway via activation of NF- κ B to protect fibroblasts from oxidative stress and excessive iron to escape ferroptosis. Interestingly, it has been reported that TNF- α /TNFR1/NF- κ B signaling could promote the activation of NOX, which is responsible for the generation of ROS (*Mediators Inflamm.* 2014;2014:279171). Since ferroptosis can be induced by accumulation of superoxide and hydrogen peroxide upon upregulation of NOX, we wondered if TNF- α /NF κ B may also activate ferroptosis-sensitizing signaling. In the revision, we showed that short-term exposure of fibroblasts to TNF- α markedly increased ROS levels, which was reduced by pretreatment with the inhibitor of NOX (NOXi) (revision Fig. 6p). However, long-term exposure of fibroblasts to TNF- α failed to result in permanent increase of ROS (revision Fig. 6p). Of note, NOXi further enhanced the protective effect of TNF- α against ferroptosis triggered by RSL3 (revision Fig. 6q). Although both TNFR1 and TNFR2 signaling have been proven to

initiate NF- κ B function via activation of TRAF-2, the increase in NF- κ B activity in response to TNFR1 activation is rapid and transient, whereas TNFR2 activation results in a much slower but persistent response (*J Biol Chem*2004;279:32869-32881). To confirm the involvement of TNFR2 in the regulation of NF- κ B activity in RA fibroblasts, we treated fibroblasts with anti-TNFR1 and anti-TNFR2 antibodies, respectively, under long-term stimulation of TNF- α . We found that blocking of TNFR2 abrogated NF- κ B activation compared to blocking of TNFR1, measured by I κ B phosphorylation (revision Fig. 4j), indicating that long-term stimulation of TNF- α tend to activate NF- κ B signaling through TNFR2 other than TNFR1. These data suggest that the defensive effect of TNF- α /TNFR2/NF- κ B against ferroptosis overcomes the ferroptosis-sensitizing contribution of TNF- α /TNFR1/NF κ B-induced ROS production.

Response Figure 13

2. Line 130: The authors should quantify the contention that surviving FAP+ fibroblasts are more likely to be surrounded by macrophages.

Response: We quantified the distance between FAP+ fibroblasts and F4/80+ macrophages using HALO™ Image Analysis Software. The number of F4/80+ macrophages within the 50 μ m radius of surviving FAP α + fibroblasts that evaded IKE-induced ferroptosis significantly increased in joints of IKE-treated CIA mice (revision Fig. 2f-g).

Response Figure 14

3. The data from Figure 2l through 2r are primarily computational and from biologic studies. Comments like “TNF signaling through NFκB was enriched in fibroblast clusters 1 and 2” , “Macrophages also initiated TGFβ signaling on fibroblasts” , “TNF released from macrophages showed a higher interaction with TNFR2 in Fib a than in Fib b” and many others are inferences rather than the result of biologic validation. The text could be significantly shortened and the comments here and throughout the manuscript should distinguish between computational predictions, potential associations based on informatics and biological validation. The authors should test some of these predictions and present data supporting them.

Response: We agree that the text could be significantly shortened and some conclusions based on single-cell RNA sequencing and assumptions should be eliminated. Thus, we deleted all comments about TGF β , PDGF and EMT signaling based on single-cell RNA sequencing. We also agree that some key predictions should be tested to provide evidence of biologic validation and to support our points of view. We tested the interaction between TNFα and TNFR1/TNFR2 as well as the activation of NFκB signaling, which will also help to explain the question #1. From single-cell RNA sequencing, we predicted the molecular interactions involved in the signaling of crucial cytokines in RA and found that fibroblasts transmitted most of the TNF-α signals via TNFR1 and TNFR2, and TNF-α provided by macrophages showed a higher interaction with TNFR2 in Fib a than in Fib b cluster (revision Fig. 4i). To confirm the involvement of TNFR2 in the regulation of NF-κB activity in RA fibroblasts, we treated fibroblasts with anti-TNFR1 and anti-TNFR2 antibodies, respectively, under long-term stimulation of TNF-α. We found that blocking of TNFR2 abrogated NF-κB activation compared to blocking of TNFR1, measured by IκB phosphorylation (revision Fig. 4j), indicating that long-term stimulation of TNF-α tent to activate NF-κB signaling through TNFR2 other than TNFR1. Thus, the stronger interaction between TNF-α from macrophages and TNFR2 in Fib a cluster is consistent with the enrichment of TNF-α signaling through NF-κB in these fibroblasts.

Response Figure 15

4. The fact that TNF, IL-1 and IL-6 had different effects on ferroptosis provides a way to dissect the signaling mechanisms. The authors conclusion that NFkB is the primary mechanism related to the TNF effect seems unlikely because IL-1 also induces the same transcription factor. A more detailed signaling study defining how TNF leads to GSH production would be important. If it is through NFkB, then methods to prevent NFkB activation would be an important step to proving their hypothesis.

Response: We thank the reviewer for this suggestion. Since the reviewer suggested to “focus studies on the key data and eliminate negative data related to the other cytokines” to make the paper more accessible and understandable, we decided to delete IL1β results. In the original version, we already used PS1145, an inhibitor of IκB kinase (IKK) that specifically inhibits IKK-mediated IκB phosphorylation to prove that TNF promotes GSH production through NFκB (original version Figure 3k-n, revision Fig. 6k-o). We found that PS1145 effectively inhibited NF-κB activation and abrogated the upregulation of *SLC7A11*, *GCLC* and *GCLM* expression (revision Fig. 6k-l). In addition, PS1145 obviously hindered TNF-α-mediated defense of ferroptosis and lipid ROS accumulation (revision Fig. 6m-n). Consistently, TNF-α-induced GSH accumulation was effectively terminated upon PS1145 treatment (revision Fig. 6o).

Response Figure 16

5. The translational approach of testing etanercept and IKE is interesting and important, although they have not shown whether the effects are due to modulating adaptive or innate immune mechanisms. For example, what was the effect on type II collagen antibodies? The authors should test their treatment regimen in established disease (typically day 35 or later in mouse CIA) or in a model that is not dependent on adaptive immunity such as a serum transfer model to determine the mechanism of action. Prolonged treatment of mice with human etanercept leads to neutralizing antibodies and occurs within a couple weeks. This issue can be avoided with shorter term studies.

Response: We thank the reviewer for bringing up this important question. CIA model requires intact adaptive immune response, particularly through the activation of pathogenic collagen-specific T cells and the generation of anti-collagen antibodies (*Springer Semin Immunopathol. 2003 Aug;25(1):3-18.*). The treatment regimen was tested in established disease (significant phenotype of arthritis) as shown in original Extended Data Fig 1c (**revision Supplementary Fig. 1f**), although not started at day 35 or later. In the revision, we tested the effect of etanercept and IKE on type II collagen antibodies in serum of CIA mice. We found that the anti-type II collagen antibody in serum of CIA mice did not differ significantly between mice treated with etanercept in combination with IKE and mice treated with vehicle (**revision Supplementary Fig. 12b**), suggesting that the reduced inflammation and joint damage are not due to modulating autoantibodies production.

As a crucial cell type of the innate immune system, macrophages play a central role in initiating and driving the pathogenesis of RA. Our data showed that the combination of etanercept with IKE not only caused ferroptotic cell death in fibroblasts, but also decreased the population of macrophages. Compared with the fibroblasts in the healthy joints that have modest immune-regulatory functions, RA fibroblasts have emerged as important immune modulators regulating the influx of the inflammatory infiltrate through secreting inflammation factors and by engaging in crosstalk with neighboring immune cells, especially macrophages. In addition to the passive reaction to pro-inflammatory stimuli from macrophages, RA fibroblasts in the synovial lining contribute actively to the pathogenesis of RA by cooperating with macrophages through direct cell-cell interactions via ligation of CD55 on fibroblasts with CD97 on macrophages (*Arthritis Rheum. 1999 Apr;42(4):650-8.*). Prostaglandins secreted by RA fibroblasts work in collaboration with pro-inflammatory factors to

shift macrophages towards a state characterized by high expression of pro-heparin-binding EGF-like growth factor (HBEGF), which further promote fibroblasts invasiveness (*Sci Transl Med.* 2019 May 8;11(491):eaau8587). RANKL expressed by RA fibroblasts can also enhance osteoclastogenesis from macrophages (*Nat Rev Rheumatol.* 2020 Jun;16(6):316-333). Thus, the ferroptosis induction seems to attenuate the interaction between fibroblasts and immune cells, which help to mitigate the inflammation and restore synovial homeostasis.

Besides, the suggestion for shortening the term for human etanercept administration in mice is very important. In our future study, we will further optimize the treatment time to avoid the influence of neutralizing antibodies.

Response Figure 17

revision Supplementary Fig. 1f

revision Supplementary Fig. 12b

6. RSL3 concentrations used in these experiments were 0.125 μ M. Others, however, require higher concentration (2 μ M, 3 μ M, and 5 μ M) than those (*Oncogene.* 2017 Oct 5;36(40) 5593-5608; *Cell Death Diff.* 2020 Dec 17; *Front Pharmacol.* 2018 Nov 22;9:1371). Prior to do in vitro cell experiment with IKE, RSL3, and others, the author should perform dose responses to determine the optimal concentrations.

Response: We thank the reviewer for this question. The dose response of fibroblasts to IKE and RSL3 was shown in original version Figure 3a (revision Fig. 5a-b), and the dose response to TNF α and other cytokines was shown in original version Extended Data Figure 6a-c (revision Supplementary Fig. 8a-c). The 0.125 μ M dose for RSL3 was chosen for following experiments because we found TNF- α could strongly protect fibroblasts from low-dose RSL3, while high doses of RSL3 still induced potent ferroptosis in the presence of TNF- α . Besides, we did find that fibroblasts are much more sensitive to ferroptosis induction than most of the epithelial tumor cells. That is consistent with our previous ferroptosis paper, which found intercellular

interactions between epithelial cells suppress ferroptosis by activating the intracellular NF2 (*Nature*, 2019;572(7769):402-406). To further prove the sensitivity of fibroblasts to ferroptosis induction, we analyzed the sensitivity of a panel of fibroblast cell lines (PG13, IMR-90, MRC5, and MH7A) to ferroptosis induction. All the tested cell lines were sensitive to 0.125 μM RSL3 induction for 14 h (Response Figure 19). These results also suggest the potential of ferroptosis therapy in other diseases related to abnormal activation of fibroblasts.

Response Figure 18

Response Figure 19

7. The manuscript should be reviewed by a statistician to assess the methods, especially Fig. 1d, e, and k and Fig. 3a.

Response: We thank the reviewer for bringing up this important suggestion. The manuscript has been reviewed by a statistician and the methods for all plots were indicated in the figure legends. For example, original version Fig. 1d and 1e (revision Fig. 1f and 1g) showed joint inflammation measured by arthritis score and paw thickness. One-way ANOVA followed by Multiple Comparisons was performed to compare the means of arthritis score or paw thickness at the end point (day 22 after disease onset). Original version Fig. 1k (revision Fig. 2d and 2e) showed cell death and lipid ROS production in circulating fibrocytes from PBMCs and synovial fibroblasts from inflamed joint fluid of RA patients treated with RSL3. Two-tailed t-test was performed to compare the means of circulating fibrocytes and synovial fibroblasts under the stimulation of RSL3. Original version Fig. 3a (revision Fig. 5a and 5b) showed relative viability of fibroblasts primed with TNF- α , IL-6 or TGF- β , followed by treatment with different concentrations of IKE or RSL3. IC₅₀ values were calculated using nonlinear regression analysis (log(inhibitor) vs. response -- Variable slope (four parameters)). We found that in human fibroblasts TNF- α conferred significant resistance to ferroptosis induced by both IKE and RSL3. The IC₅₀ of IKE was 0.65 μ M in fibroblasts, 1.16 μ M in TNF- α -treated fibroblasts, 0.42 μ M in TGF- β -treated fibroblasts and 0.31 μ M in IL-6-treated fibroblasts at 26 h. The IC₅₀ of RSL3 was 0.042 μ M in fibroblasts, 0.252 μ M in TNF- α -treated fibroblasts, 0.045 μ M in TGF- β -treated fibroblasts and 0.007 μ M in IL-6-treated fibroblasts at 12 h.

Response Figure 20

8. Minor points: in line 110, the authors comment on FAP⁺ fibroblasts in RA synovium and cultured RA synovial fibroblasts. As noted by several authors, cultured SF are almost all FAP⁺/CD90⁺/podoplanin⁺ and have a phenotype that is a combination of lining and sublining fibroblasts. Also, the title should be modified because the authors do not actually test their hypothesis in patients with RA.

Response: We agree that the FAP⁺ phenotype of cultured SFs can't represent the expression of FAP in RA fibroblasts *in vivo*. Thus, we deleted the original extended Fig. 1g and modified the comment about FAP⁺ fibroblasts as follows: "Fibroblast activation protein- α (FAP α) is a widely accepted marker for fibroblasts in RA; these cells are located in both the inner and outer layers of the synovium (*Nature* 2019,570:246-251)". We also agree it's inappropriate to use the former title with the description "therapy" since we didn't test the hypothesis in patients. Thus, we modified the title of the paper to "TNF- α Antagonist Sensitizes Synovial Fibroblasts to Ferroptotic Cell Death in Rheumatoid Arthritis".

REVIEWERS' COMMENTS

Reviewer #1 (Remarks to the Author):

The authors have addressed my queries and included additional experimental data.

Reviewer #2 (Remarks to the Author):

The authors have made major improvements to the manuscript. I recommend the revised manuscript for publication in Nature Communications.